# A transcription factor collective defines the HSN serotonergic neuron regulatory landscape

**Carla Lloret-Fernández[1][†], Miren Maicas[1][†], Carlos Mora-Martínez[1][†], Alejandro Artacho[2], Ángela Jimeno-Martín[1], Laura Chirivella[1], Peter Weinberg[3], Nuria Flames[1]***

[1]Developmental Neurobiology Unit, Instituto de Biomedicina de Valencia IBV-CSIC, Valencia, Spain; [2]Departamento de Genómica y Salud, Centro Superior de Investigación en Salud Pública, FISABIO, Valencia, Spain; [3]Department of Biological Sciences, Howard Hughes Medical Institute, Columbia University Medical Center, New York, United States

**Abstract** Cell differentiation is controlled by individual transcription factors (TFs) that together activate a selection of enhancers in specific cell types. How these combinations of TFs identify and activate their target sequences remains poorly understood. Here, we identify the *cis*-regulatory transcriptional code that controls the differentiation of serotonergic HSN neurons in *Caenorhabditis elegans*. Activation of the HSN transcriptome is directly orchestrated by a collective of six TFs. Binding site clusters for this TF collective form a regulatory signature that is sufficient for de novo identification of HSN neuron functional enhancers. Among *C. elegans* neurons, the HSN transcriptome most closely resembles that of mouse serotonergic neurons. Mouse orthologs of the HSN TF collective also regulate serotonergic differentiation and can functionally substitute for their worm counterparts which suggests deep homology. Our results identify rules governing the regulatory landscape of a critically important neuronal type in two species separated by over 700 million years.
DOI: https://doi.org/10.7554/eLife.32785.001

**\*For correspondence:**
nflames@ibv.csic.es

[†]These authors contributed equally to this work

**Competing interests:** The authors declare that no competing interests exist.

## Introduction

Cell identities are characterized by the expression of specific transcriptomes that are activated through cell-type-specific regulatory landscapes. Large efforts have been made to identify functional enhancers in different tissues and developmental stages. The approaches include the occupancy of combinations of transcription factors (TFs), identifying DNA regions displaying open chromatin states, analyzing specific histone marks and assessing enhancer function by transgenesis in vivo (*Junion et al., 2012*; *Mo et al., 2015*; *Nord et al., 2013*; *Pattabiraman et al., 2014*; *Visel et al., 2013*; *Zinzen et al., 2009*). These studies have revealed a highly dynamic organization of active enhancers that change depending on the cell type and developmental stage. However, to date, it is unclear what features of the DNA sequences distinguish enhancer regions from the rest of the genome. The identification of such features is critical both for understanding fundamental biological processes such as cell fate specification, as well as for biomedicine, given that most disease-associated mutations are thought to be located within regulatory sequences (*Mathelier et al., 2015*; *Nishizaki and Boyle, 2017*).

TFs are the main regulators of enhancer function. Each enhancer is bound by specific combinations of TFs that will either activate or repress transcription (*Reiter et al., 2017*). The distribution of TF-binding sites (TFBS) has been studied in detail in only a few enhancers. For example, a study of

**eLife digest** All cells in the body essentially share the same DNA, despite looking very different and playing a range of roles. The reason that cell types are so different from one another is because of the way they interpret the DNA. Each different type of cell uses a specific subset of the genes within the genome. The part of the DNA that controls which cell will use which genes and when is called the regulatory genome; this DNA is not translated into proteins.

The regulatory genome is much less well understood than the protein-coding genome. At present, when a new species is discovered, it is often possible to sequence its DNA and deduce where the protein-coding genes are and what roles they might play. However, it is not yet possible to do the same for the regulatory genome. Finding a way to do this is an important step towards understanding when and where each of the organism's genes is active.

Lloret-Fernández, Maicas, Mora-Martínez et al. focused on the regulatory genome of nerve cells that use a chemical messenger called serotonin in the nematode worm *Caenorhabditis elegans*. First, they studied mutations in six genes that code for transcription factors that are active in this cell type. Transcription factors are proteins that identify and bind to specific regions of the genome to control the activity of nearby genes. These six mutants failed to correctly activate the regulatory genome of this nerve cell, which was measured using a genetic approach that caused the nerves to glow green under a microscope when the regulatory genome was active.

Further experiments then confirmed that all six transcription factors must act together to identify and activate the regulatory genome in this particular nerve cell. The fact that the DNA sites that these transcription factors bind are clustered close to each other means they can be used as a marker to help decode the active regulatory genome of this class of nerve cell.

This is a small step towards understanding how the regulatory genome works. Comparisons with similar nerve cells from mammals found that the equivalent transcription factors have the same role, suggesting that they may be broadly conserved across species. Understanding the regulatory genome better could eventually lead to new treatments for certain genetic conditions, as many mutations associated with diseases appear outside the protein-coding genome.

DOI: https://doi.org/10.7554/eLife.32785.002

*sparkling*, a specific enhancer of the *Drosophila* Pax2 gene, revealed it to be densely packed with TFBS that required specific arrangements for its functionality (*Swanson et al., 2010*). However, these one-by-one approaches are not able to reveal any general molecular logic underlying cell-type-specific regulatory landscapes. Chromatin immunoprecipitation combined with deep sequencing (ChIP-seq) has been used to generate genome-scale binding profiles of specific TFs. It is now clear that TF-binding profiles are dynamic during cell differentiation and vary in related species (*Garber et al., 2012*; *Heinz et al., 2010*; *Khoueiry et al., 2017*; *Nord et al., 2013*; *Stefflova et al., 2013*; *Wilczyński and Furlong, 2010*; *Zinzen et al., 2009*). However, it is unclear what distinguishes TFBS actually bound by the TF from those that are unoccupied. Moreover, despite the fact that only a small fraction of bound TFBS are located in enhancers (*Kwasnieski et al., 2014*; *Pattabiraman et al., 2014*; *Whitfield et al., 2012*), the molecular organization that distinguishes functional enhancers from the rest of non-coding regions is still unknown. Collective binding of several TFs is emerging as an important feature that distinguishes TFBS at functional enhancers from other genomic regions bound by individual TFs (*Junion et al., 2012*; *Khoueiry et al., 2017*; *Mazzoni et al., 2013*; *Zinzen et al., 2009*). However, it is still unclear how these combinations of TFs are collectively recruited to and activate cell-type-specific regulatory landscapes.

The study of the transcriptional regulatory mechanisms underlying neuronal subtype specification in vivo in complex model organisms, such as rodents, is a challenging task. Here, we take advantage of the simple model organism *C. elegans* to study neuron type specification in vivo. *C. elegans* is especially suitable for transcriptional regulatory studies because its cell lineage is fully described, it is easy to genetically manipulate and its genome is very compact (despite containing a similar number of genes to the human genome) (*Gerstein et al., 2010*). In this work, we focus on the study of the transcriptional regulatory logic of serotonergic neurons. Serotonergic neurons are present in all eumetazoan groups and are universally defined by their ability to synthesize and release serotonin

(5HT) (*Flames and Hobert, 2011*). They regulate multiple processes and their dysfunction has been linked to bipolar disorder, depression, anxiety, anorexia and schizophrenia (*Deneris and Wyler, 2012*; *Mathelier et al., 2015*). Several TFs are known to be involved in mammalian serotonergic differentiation (*Deneris and Wyler, 2012*). However, little is known about their function in the regulation of specific serotonergic neuron enhancers. Here, we focus on this clinically relevant and highly conserved neuronal subtype, and exploit the amenability of *C. elegans* to unravel the rules governing the activation of the serotonergic transcriptome. This work reveals the phylogenetically conserved action of a collection of TFs on the selection of a specific regulatory landscape from the genome and allows for the identification of neuron subtype specific functional enhancers merely based on the presence of the TF collective regulatory signature.

## Results

### Transcription factors from six different families are required for HSN neuron terminal differentiation

Serotonergic neurons are characterized by the coordinated expression of a battery of phylogenetically conserved enzymes and transporters known as the 5HT pathway genes (*Figure 1A*). *C. elegans* adult hermaphrodites contain three functionally distinct serotonergic neuron subclasses: the NSM neurosecretory neuron, the ADF chemosensory neuron and the HSN motor neuron (*Figure 1B*), which arise from different progenitors and, with the exception of the 5HT pathway genes, express different effector genes (*Figure 1—figure supplement 1*). The HSN neuron is, by far, the best characterized, thus we focused on dissecting the transcriptional rules governing the differentiation of this subclass.

The HSN motor neuron controls vulval muscle contraction and its dysfunction leads to an egg laying defective (*egl*) phenotype. To identify the TF combination controlling HSN terminal differentiation, we selected among previously described *egl* mutants, those that both code for TFs and had reduced or no staining of 5HT in HSN. At least four genes matched this criteria: the POU domain TF *unc-86*, the Spalt-type Zn finger TF *sem-4*, the bHLH domain TF *hlh-3* and the Insm-type Zn finger TF *egl-46* (*Basson and Horvitz, 1996*; *Doonan et al., 2008*; *Sze et al., 2002*; *Wu et al., 2001*). In addition to previous reports, we found that the GATA factor *egl-18*, a regulator of HSN migration (*Desai et al., 1988*) and the ETS TF *ast-1*, a regulator of dopaminergic fate (*Flames and Hobert, 2009*), also exhibit an HSN 5HT staining phenotype not previously published. Thus, although additional TFs with subtler *egl* phenotypes or with pleiotropic lethal effects and no available hypomorphic alleles are likely to be required for correct HSN differentiation, we initially focused our study in this set of six TFs that we refer as the HSN TF combination.

To confirm previous observations, we analyzed null alleles for each member of the HSN TF combination except for *ast-1*, where we used a hypomorphic allele as the null allele is lethal prior to HSN differentiation (*Schmid et al., 2006*). All TF mutants indeed displayed a defective egg laying phenotype and 5HT staining and 5HT pathway gene expression defects, further supporting their roles in HSN differentiation (*Figure 1C,E* and *Source data 1*). We observed similar 5HT staining and 5HT pathway gene expression defects by RNAi knock down and in the analysis of additional mutant alleles for each candidate TF confirming that the phenotype is due to mutations in each corresponding TF and not to background strain effects (*Figure 1—figure supplement 2* and *Supplementary file 1*). Importantly, 5HT pathway gene expression defects were specific for the HSN serotonergic subclass, while ADF and NSM neurons were unaffected (*Source data 1*), the exception being *unc-86(n846)* which showed a previously reported NSM differentiation phenotype (*Sze et al., 2002*; *Zhang et al., 2014*).

We next assessed whether the HSN TF combination is also required for the expression of non-5HT related genes by analyzing nine additional reporters. We observed expression defects in all mutant strains (*Figure 1D,E* and *Source data 1*). Although most terminal features were affected, the expression of some genes remained normal indicating that, in each single mutant, HSN neuron is present and shows broad but partial differentiation defects (*Figure 1D,E*). HSN neuron is born embryonically and remains in a quiescent undifferentiated state until fourth larval stage (L4), when it activates the expression of most effector genes, including the 5HT pathway genes. We did not observe precocious expression of any of the analyzed terminal features in any of the single mutant

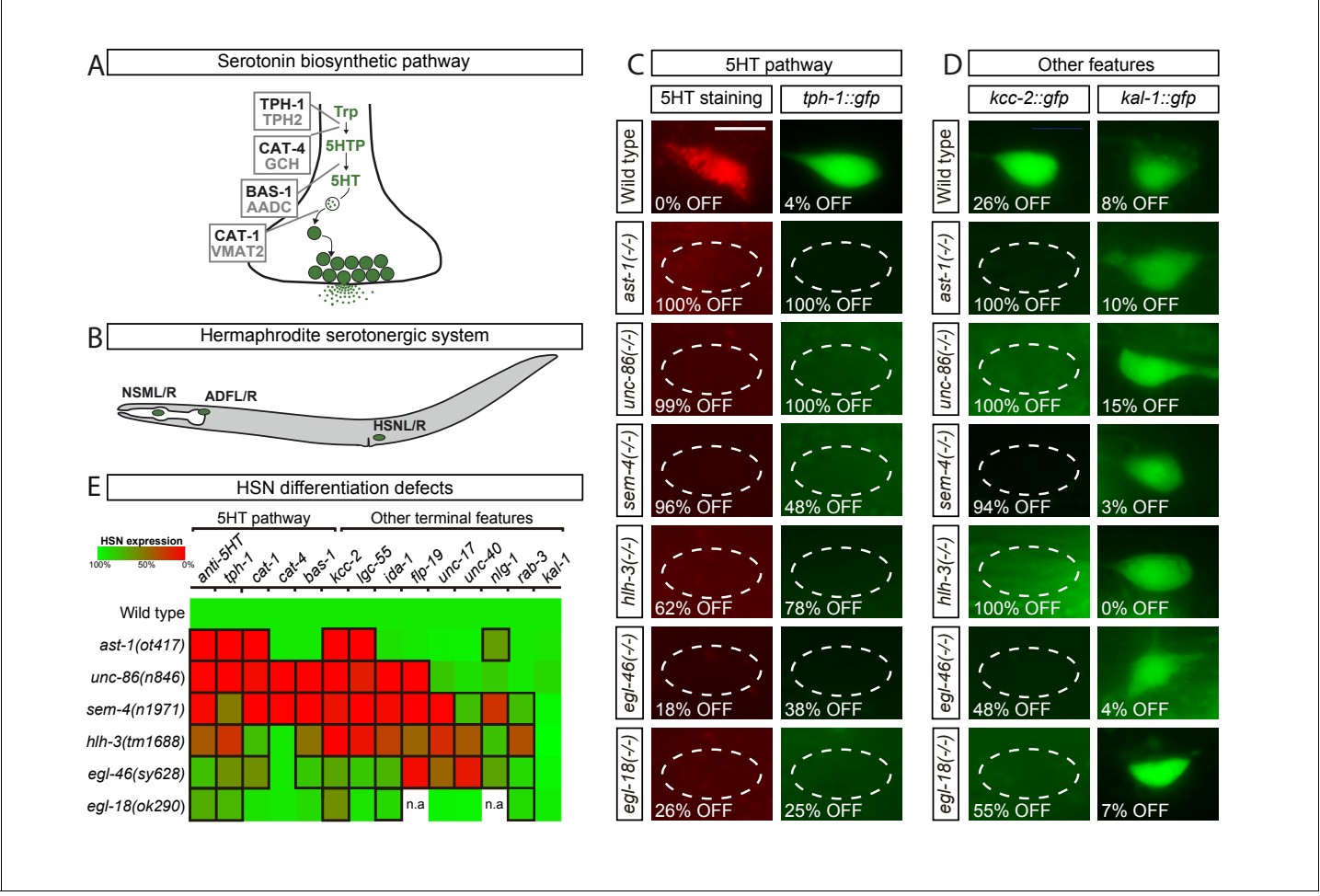

**Figure 1.** Transcription factors from six different TF families are required for HSN terminal differentiation. (**A**) Phylogenetically conserved serotonin biosynthetic pathway. *C. elegans* protein names appear in black case, mammalian in grey. AADC: aromatic L-amino acid decarboxylase; GCH: GTP cyclohydrolase; TPH: tryptophan hydroxylase; Trp: tryptophan; VMAT: vesicular monoamine transporter; 5HTP: 5-hydroxytryptophan; 5HT: serotonin. (**B**) *C. elegans* hermaphrodite serotonergic system is composed of three subclasses of bilateral neurons (NSM, ADF and HSN, L: left, R: right). See *Figure 1—figure supplement 1* for expression profiles of serotonergic subclasses. (**C**) Micrographs showing HSN 5HT staining and *tph-1::gfp* reporter expression defects of *ast-1(ot417)*, *unc-86(n846)*, *sem-4(n1971)*, *hlh-3(tm1688)*, *egl-46(sy628)* and *egl-18(ok290)* mutant animals (quantified in E). Scale bar: 5 μm. (**D**) Micrographs showing expression defects in the K$^+$/Cl$^-$ cotransporter *kcc-2::gfp* reporter, a terminal feature of HSN not related to 5HT signaling, and normal expression of the extracellular matrix gene *kal-1*, indicating HSN is still present. (**E**) Heatmap summary of single TF mutant characterization. Statistically significant expression defects compared to wild type are indicated with a black frame. *flp-19*: FMRF-like peptide; *ida-1*: Tyr phosphatase-like receptor; *lgc-55*: amine-gated Cl$^-$ channel; *nlg-1*: neuroligin; *rab-3*: ras GTPase; *unc-17*: vesicular acetylcholine transporter; *unc-40*: netrin receptor. n.a: not analyzed. See *Source data 1* for primary data and Fisher's exact test p-values and *Figure 1—figure supplement 2* and *Supplementary file 1* for analysis of additional alleles. n > 100 cells per condition.

DOI: https://doi.org/10.7554/eLife.32785.003

The following figure supplements are available for figure 1:

**Figure supplement 1.** Each serotonergic neuron subclass expresses different sets of genes.
DOI: https://doi.org/10.7554/eLife.32785.004

**Figure supplement 2.** Schematic representation of analyzed HSN TF combination alleles.
DOI: https://doi.org/10.7554/eLife.32785.005

backgrounds, suggesting that the HSN TF combination acts mainly as activator of transcription. Most notably, the phenotypic profile of each mutant was slightly different from the others, which suggests that these TFs do not function in a cascade-like linear pathway (*Figure 1E*).

## AST-1 acts as temporal switch for HSN maturation

The expression of the HSN TF combination has only been partially studied (*Basson and Horvitz, 1996*; *Doonan et al., 2008*; *Finney et al., 1988*; *Wu et al., 2001*). We used fosmid reporter strains (for *unc-86*, *sem-4* and *egl-18*), endogenous locus tagging (for *ast-1* and *hlh-3*, *Figure 2—figure supplement 1*) and a transcriptional reporter strain (for *egl-46*) to analyze their expression pattern in HSN throughout development. We find that all six TFs are expressed in the HSN at L4 coinciding with the onset of differentiation (*Figure 2A*). The HSN TF combination is also expressed in other neurons, including expression of UNC-86, EGL-18 and EGL-46 in the NSM serotonergic neuron, while none of them are expressed in the ADF serotonergic neuron.

A deeper analysis of the developmental expression of each TF shows a very diverse array of expression dynamics (*Figure 2B*). Some HSN TFs are expressed embryonically (such as UNC-86, HLH-3 or EGL18). SEM-4 is widely expressed in the embryo in the area were HSN is located, although it is likely to be expressed at this early stage in HSN, we could not unequivocally identify it. In contrast, AST-1 and EGL-46 initiate their expression at different postnatal stages (*Figure 2B*). Interestingly, HLH-3 shows two waves of expression: it is present in the mother cell of HSN (around 280 min of embryonic development, see lineage in *Figure 2—figure supplement 1*) and its expression becomes fainter in postmitotic HSN and PHB neurons (data not shown). At first larval stage (L1), HLH-3 expression is undetectable in HSN and expression reappears at third larval stage (L3), preceding AST-1 onset of expression and HSN maturation and is quickly downregulated at the end of L4 [*Figure 2B*, *Figure 2—figure supplement 1* and (*Doonan et al., 2008*)].

Thus, to further study the temporal requirements of HLH-3 activity in HSN differentiation, we induced HLH-3 expression at early L4 state in *hlh-3* mutants. This late expression is sufficient to rescue *tph-1* reporter expression defects indicating that embryonic HLH-3 expression is not required for correct HSN terminal differentiation (*Figure 2C*).

Little is known about the temporal control of HSN differentiation. Heterochronic genes have been described to regulate the onset of HSN axon extension (*Olsson-Carter and Slack, 2010*), although the molecular mechanisms underlying this process are unknown. AST-1 and HLH-3 expression correlates with HSN maturation suggesting they might have a role in determining the onset of this process. We used an early active HSN promoter to induce *ast-1* and *hlh-3* expression precociously from first larval stage (L1). Our results show that AST-1 significantly advances *tph-1* expression to L2-L3 larval stages (*Figure 2D*). On the contrary, early *hlh-3* induced expression either alone or in combination with *ast-1* leads to both a delay in onset and expression defects of *tph-1* reporter gene (*Figure 2D*). Despite *tph-1* expression defects, HSN was still present as we could identify it by differential interference contrast (DIC) (data not shown). These results suggest that AST-1 activity is an important determinant of HSN maturation onset. We also found that *lin-41* heterochronic mutants show *ast-1* expression defects in the HSN (data not shown) further supporting the role of AST-1 as a downstream effector controlling HSN maturation timing.

Additionally, these experiments underscored the importance of the dynamic regulation of *hlh-3* expression. HLH-3 is a proneural TF of the asc family, ortholog of mouse Ascl1 and *Drosophila* Scute (*Figure 2—figure supplement 1*). Proneural factors regulate both neural progenitor specification and neuronal differentiation and their functions are conserved through evolution from cnidarians to mammals (*Guillemot and Hassan, 2017*). Ascl1 is required for correct mouse serotonergic specification (*Pattyn et al., 2004*) and its activity is required to induce serotonergic fate from human fibroblasts (*Vadodaria et al., 2016*; *Xu et al., 2016*). HLH-3 shows several features common to ASCL1: (1) HLH-3 is transiently expressed in all neuronal progenitors and differentiating neuroblasts and it is required for correct differentiation of several neuronal types, including HSN (*Doonan et al., 2008*; *Gruner et al., 2016*; *Krause et al., 1997*; *Luo and Horvitz, 2017*; *Murgan et al., 2015*). (2) Both HLH-3 and ASCL1 are required to induce correct neurotransmitter identity (*Pattyn et al., 2004*; *Sommer et al., 1995*). (3) As will be explained in a later section, Ascl1 can rescue HSN differentiation defects of *hlh-3* mutants, supporting its functional conservation. (4) As would be expected for a proneural gene, HLH-3 is also required for correct expression of panneuronal features in the HSN (*Figure 1*, *rab-3* expression defects). (5) We find that HLH-3 expression needs to be tightly temporally regulated to correctly induce HSN fate. Temporal regulation of ASCL1 and SCUTE activities is also required for correct neuronal specification (*Andersen et al., 2014*; *Imayoshi et al., 2013*; *Quan et al., 2016*; *Urbán et al., 2016*). (6) HLH-3 regulates *egl-46* expression (discussed in next

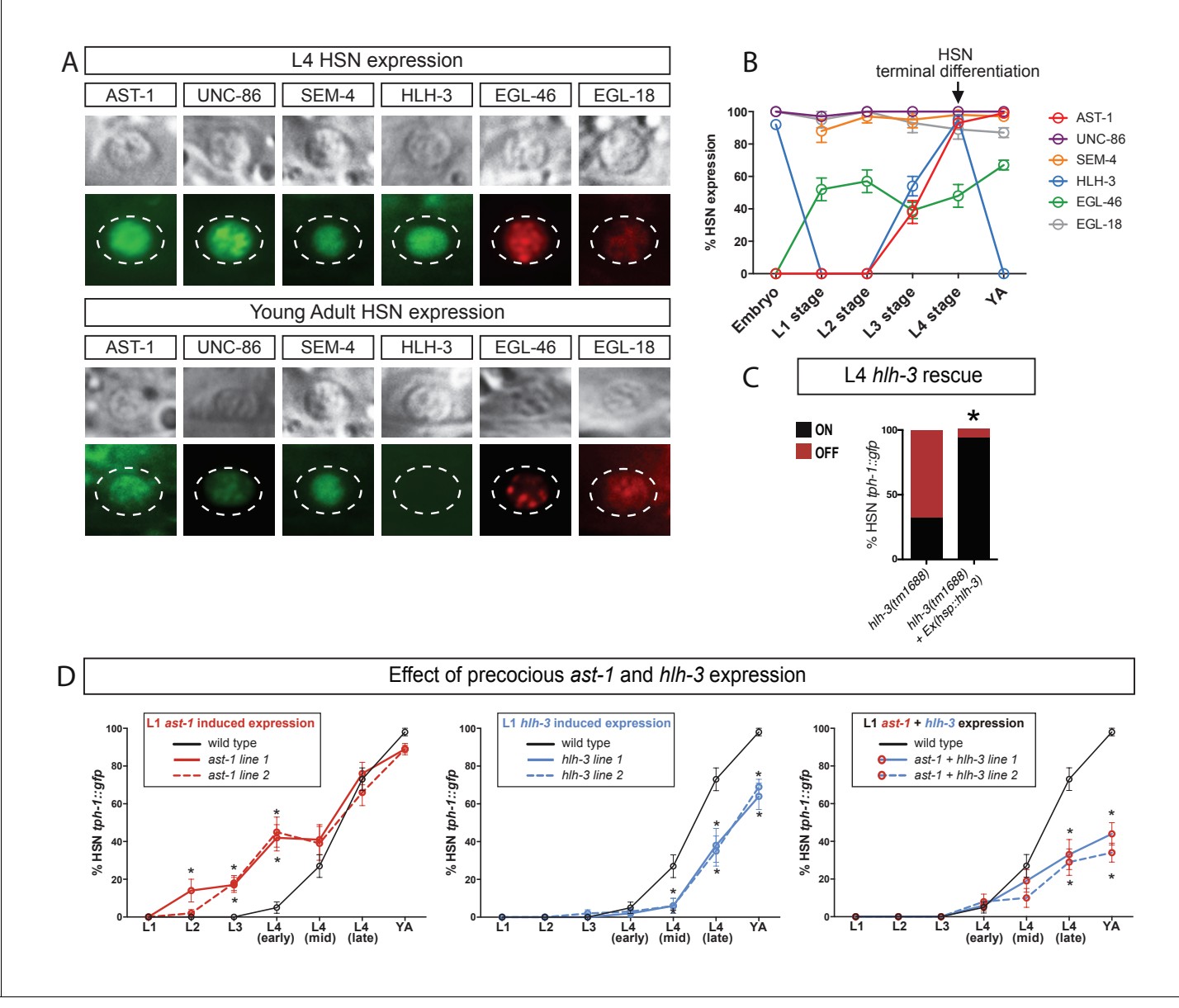

**Figure 2.** AST-1 acts as temporal switch for HSN maturation. (**A**) Micrographs showing expression of the HSN TF combination at L4 larval stage and adult animals. (**B**) Analysis of HSN TF expression across all developmental stages in the HSN neuron. n > 30 cells for each developmental point. Error bars are SEP values. See *Figure 2—figure supplement 1* for more detailed *hlh-3* developmental expression. (**C**) Heat-shock-induced expression of *hlh-3* at L4 larval stage is able to rescue *tph-1::gfp* expression defects in the HSN neuron. n > 100 cells per condition. See *Source data 1* for primary data and Fisher's exact test p-values. *: p-value <0.05. (**D**) Precocious L1 onset of expression of *ast-1, hlh-3* or both using an early active HSN-specific promoter (also expressed in NSM, ADF and VC4/5 neurons). Precocious *ast-1* advances *tph-1::gfp* expression, while *hlh-3* alone or in combination with *ast-1* delays *tph-1::gfp* expression and produces expression defects. YA: young adult. n > 30 cells per time point and condition. See *Source data 1* for primary data and Fisher's exact test p-values. *: p-value <0.05.

DOI: https://doi.org/10.7554/eLife.32785.006

The following figure supplement is available for figure 2:

**Figure supplement 1.** Dynamic HLH-3 expression in the HSN.

DOI: https://doi.org/10.7554/eLife.32785.007

section, *Figure 3A*) and similarly, Insm1 (ortholog of *egl-46*) is a direct target of Ascl1 (*Castro et al., 2011*). Taken together, our data suggests that HLH-3 acts as a proneural factor in HSN specification. Future experiments will determine if, similar to ASCL1 (*Wapinski et al., 2013*), HLH-3 acts as a pioneer factor to facilitate binding of other TFs and promote neural differentiation.

## UNC-86 is a master regulator of the HSN transcription factor combination

We next examined whether these TFs could exhibit cross regulation by analyzing their expression in each of the six different TF mutant backgrounds. In most cases, expression of each TF was largely independent of the integrity of the rest of the HSN TFs (*Figure 3A* and *Source data 1*). However, UNC-86 is a notable exception as it is required for the expression of most factors (*Figure 3A*). Noteworthy, SEM-4, that is downstream UNC-86, is also required for AST-1 and partially for HLH-3 expression. Thus, UNC-86 effects could be, at least in part due to SEM-4 regulation. Finally, additional more modest effects are also observed between other TF pairs, such as the regulation of AST-1 and EGL-46 by HLH-3 (*Figure 3A* and summarized in *Figure 3B*).

Since TFs required for neuronal terminal differentiation are often also required to maintain the correct differentiated state (*Deneris and Hobert, 2014*), we explored whether this was also the case for the HSN TFs. We find that UNC-86, SEM-4, AST-1, EGL-46 and EGL-18 expression is maintained in HSN after differentiation while HLH-3 expression is not observed after larval L4 stage (*Figure 2A and B*). RNAi experiments to knock down the expression of the adult expressed TFs after HSN maturation produce defects in the maintenance of *tph-1* and *cat-1* expression (*Figure 3C*). Additionally, the use of temperature-sensitive alleles for *ast-1*, *unc-86* and *sem-4* leads to similar maintenance defects (*Figure 3—figure supplement 1*). Our results revealed that these five TFs are continuously required to maintain the correct HSN differentiated state.

## Distinct *cis*-regulatory modules control serotonin pathway gene expression in different subclasses of serotonergic neurons

We next performed a comprehensive, in vivo analysis of the *cis*-regulatory modules (CRMs) for 5HT pathway genes to analyze how the HSN TFs regulate HSN terminal differentiation.

First, we dissected the regulatory regions of the 5HT pathway genes by in vivo reporter analysis and isolated the minimal CRMs able to direct expression in each serotonergic neuron subclass (*Figure 4* and *Figure 4—figure supplement 1*). We found that for each gene, different CRMs were active in specific subclasses of serotonergic neurons (HSN, NSM or ADF). This suggests that expression of the same 5HT pathway gene is independently regulated in each of the three serotonergic neuron subclasses. These results, together with previous reports of different TF mutants affecting specific subclasses of serotonergic neurons (*Desai et al., 1988*, *Olsson-Carter and Slack, 2010*, *Zhang et al., 2014*, *Zheng et al., 2005*) support the presence of subclass-specific serotonergic differentiation programs. Of note, 5HT pathway gene CRMs in some cases partially overlap (*Figure 4*). This overlap might be due to the presence of shared TFBS among serotonergic neuron subclasses and indeed, UNC-86 regulates both NSM and HSN differentiation (*Sze et al., 2002*). We found that disruption of POU TFBS in *tph-1* and *bas-1* HSN CRMs but not in *cat-1* CRM (discussed in the following section) affects both HSN and NSM expression. TFs are pleiotropic and it is known that the same TF can act with different combinations of TFs in different neuronal types to control neuron-type-specific genetic programs (*Hobert, 2016*).

The observed serotonergic subclass independent regulation of 5HT pathway genes is in sharp contrast with our previous study of the dopaminergic regulatory logic in which all four subclasses of dopaminergic neurons are regulated by the same combination of TFs and through unique CRMs (*Doitsidou et al., 2013*; *Flames and Hobert, 2009*). Dopaminergic neuron subclasses are functionally similar (mechanosensory neurons), which may explain why they can share a unique TF combination to select a similar transcriptome. Conversely, the functional and molecular diversity of serotonergic neuron subclasses would require independent TF programs to select diverse terminal transcriptomes.

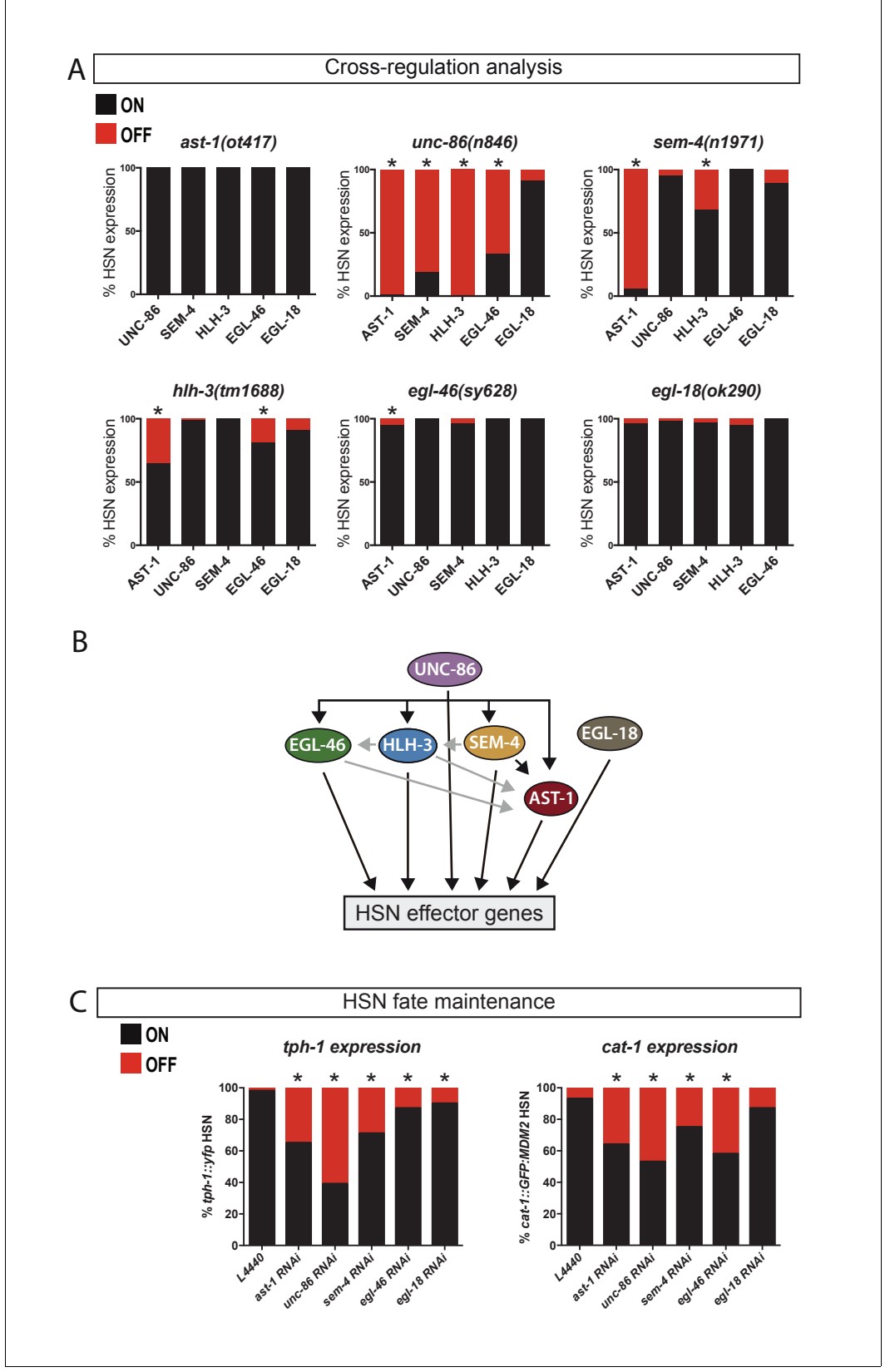

**Figure 3.** UNC-86 is a master regulator of the HSN transcription factor combination. (**A**) Expression of the HSN TFs in different mutant backgrounds. All scorings were performed at adult stages except for HLH-3, where early L4 larvae were scored. Embryonic HLH-3 expression is unaffected in *unc-86* mutants (data not shown). Graphs show *Figure 3 continued on next page*

*Figure 3 continued*

the percentage of TF expression in mutant animals relative to wild type expression. n > 100 cells per condition, Fisher's exact test, *: p-value<0.05, See *Source data 1* for raw data and exact p-values. (B) Summary of relationships among the HSN TF combination, black arrows mean strong effect (more than 50% loss of expression) and grey arrows depicts the rest of significant defects. (C) Loss-of-function (RNAi) experiments after HSN differentiation show that AST-1, UNC-86, SEM-4, EGL-46 and EGL-18 are required to maintain proper *tph-1::yfp* and *cat-1::MDM2::gfp* (unstable GFP) reporter expression. Worms were also scored prior to RNAi treatment to confirm correct HSN differentiation before starting the experiment. n > 100 cells per condition, Fisher's exact test, *: p-value <0.05. See *Source data 1* for raw data and *Figure 3—figure supplement 1* for maintenance analysis with temperature-sensitive alleles.

DOI: https://doi.org/10.7554/eLife.32785.008

The following figure supplement is available for figure 3:

**Figure supplement 1.** AST-1, UNC-86 and SEM-4 are required to maintain the HSN differentiated state.

DOI: https://doi.org/10.7554/eLife.32785.009

## HSN transcription factor combination acts directly on target genes

Next, to assess whether the action of the HSN TF combination was direct on the serotonergic regulatory regions, we focused our analyses on the HSN minimal CRMs from the three 5HT pathway genes that showed the strongest phenotypes in our previous mutant analysis: *tph-1* (TPH), *cat-1* (VMAT) and *bas-1* (AADC). We performed site-directed mutagenesis on predicted TFBS in these CRMs and analyzed in vivo the effect of the mutations.

Our analysis, explained in detail below, revealed that all members of the HSN TF combination act directly upon 5HT pathway gene CRMs. Each CRM has a different disposition of TFBS arrangements supporting the flexible function of the HSN TFs. Additionally, we found examples of redundancy between TFBS that provide robustness of expression to the system and whose functionality can only be revealed in the context of smaller CRMs or mutant backgrounds. Notably, redundancy is specific to the CRM architecture as two TFs can act redundantly in one CRM but not in others. Finally, we also found that short HSN CRMs that lack TFBS for some HSN TF members can drive partially penetrant HSN expression, while longer CRMs with functional binding sites for additional members of HSN TFs drive more robust expression. This direct but flexible action of a combination of TFs to directly regulate cell type specification has been previously termed 'TF collective' mode of regulation (*Junion et al., 2012*; *Spitz and Furlong, 2012*), accordingly, we termed this set of TFs the 'HSN TF collective'.

The HSN minimal CRM for *tph-1* (TPH) (*tph-1prom2*, *Figure 4A*) contained predicted binding sites for all six HSN TF members (*Figure 5A*). In vivo mutation reporter analyses revealed that all except the SPALT- and GATA-binding sites were required for proper *tph-1* expression in HSN (*Figure 5A* and *Figure 5—figure supplement 1*). SEM-4 (SPALT) is required for *ast-1* expression thus its effect on *tph-1* expression could be indirect. Paradoxically, *egl-18* (GATA) mutants showed defects in *tph-1prom2* expression, similar to what was observed for the full-length reporter (*Figure 5B*). Taking into account that EGL-18 does not regulate the expression of any member of the HSN TF collective, it may act upstream of another unidentified TF to regulate *tph-1prom2* expression. Alternatively, EGL-18 may be recruited to the *tph-1* promoter even in the absence of functional GATA -binding sites, perhaps through interactions with other members of the HSN TF collective. Similar binding site-independent recruitment of TFs, when combinatorially binding in a TF collective, has been reported for other combinations of TFs (*Junion et al., 2012*; *Uhl et al., 2016*).

The HSN minimal CRM for *cat-1* (VMAT) (*cat-1prom14*, *Figure 4B*) also contained predicted binding sites for the HSN TF collective (*Figure 5C*). Point mutation analyses revealed functionality of all but INSM-binding sites (*Figure 5C* and *Figure 5—figure supplement 1*). In agreement with this observation, we found that *cat-1prom14* expression does not require EGL-46 (INSM) factor (*Figure 5D*). However, the penetrance of HSN expression for the minimal *cat-1prom14* was much lower than for the full-length reporter (55% versus 100% expression, respectively, *Figure 5D*). This indicates that additional TFBS outside of the minimal CRM are required to promote robust HSN expression. Indeed, full-length reporter (*cat-1prom1*) expression was affected in *egl-46* mutants (*Figure 5D*). These results suggest that, although partial expression from *cat-1* can be achieved without EGL-46, this TF is required for robust expression in the context of the full *cat-1* promoter.

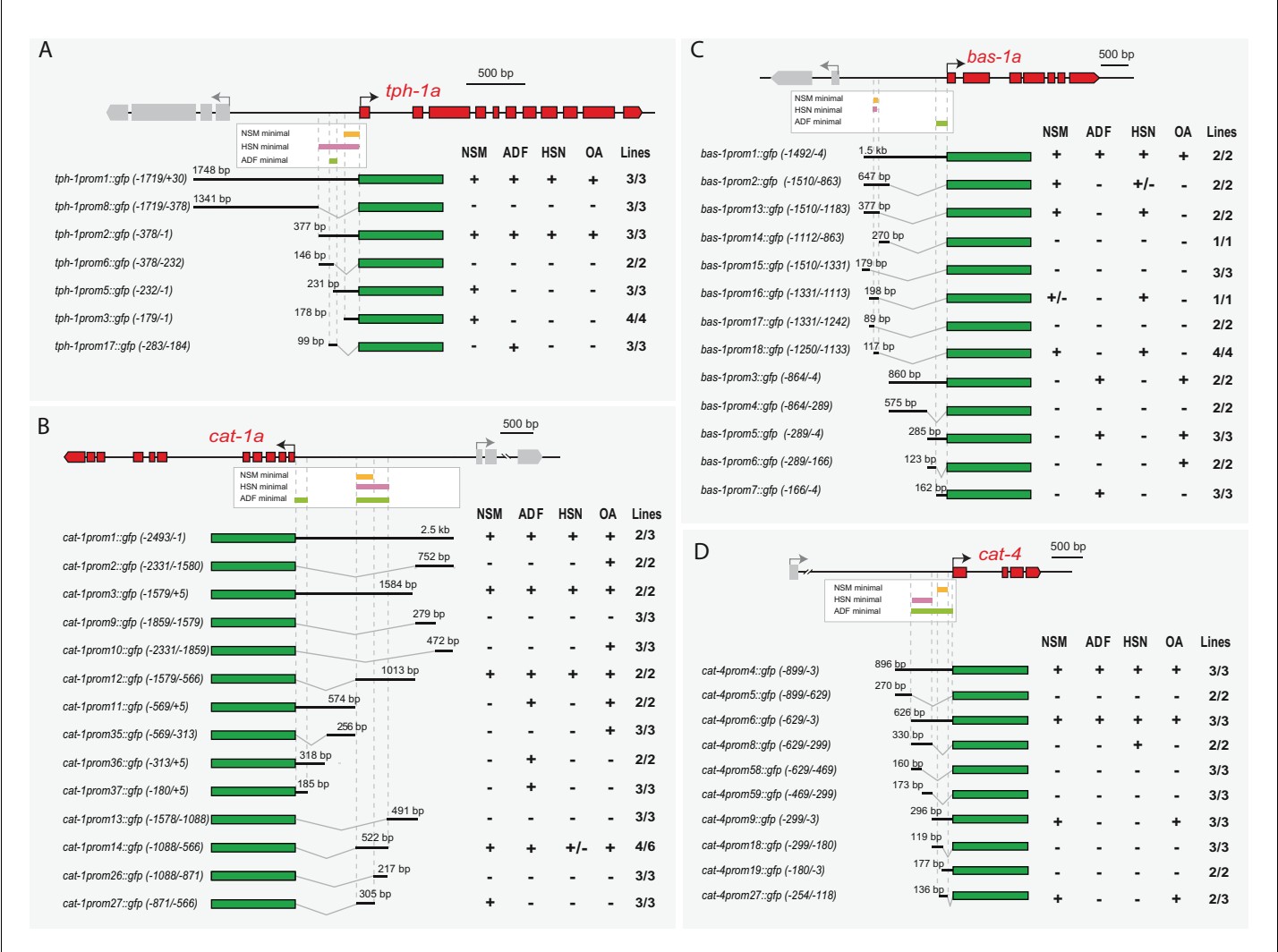

**Figure 4.** Distinct *cis*-regulatory modules control serotonin pathway gene expression in different subclasses of serotonergic neurons (A–D) *cis*-regulatory analysis of the 5HT pathway genes. White boxes underneath each gene summarize the smallest CRM that drive expression in each serotonergic neuron subclass. Thick black lines symbolize the genomic region placed upstream of GFP (green box) and dashed lines are used to place each construct in the context of the locus. OA: other aminergic cells (RIC, RIM, AIM, RIH, CEPs, ADE, PDE, VC4/5) that also share the expression of some 5HT pathway genes. Numbers in brackets represent the coordinates of each construct referred to the ATG. +: >60% GFP positive cells; +/−: 20–60% GFP cells; −: <20% GFP cells. x/y represents the number of lines with the expression pattern (x) from the total lines analyzed (y). n > 60 cells per line. See *Figure 4—figure supplement 1* for raw values.

DOI: https://doi.org/10.7554/eLife.32785.010

The following figure supplement is available for figure 4:

**Figure supplement 1.** 5HT pathway gene CRM analysis.
DOI: https://doi.org/10.7554/eLife.32785.011

The requirement for GATA sites in the *cat-1* minimal CRM contrasted with the lack of an expression defect of a full-length *cat-1* reporter in *egl-18* (GATA) mutants (*Figure 5D*). However, when we analyzed the minimal *cat-1* CRM (*cat-1prom14*) in *egl-18* mutants we found that its activity was affected in this mutant background (*Figure 5D*). Thus, EGL-18 directly regulates *cat-1* expression but its loss can be compensated in the context of a large regulatory region by other unknown factors. We confirmed EGL-18 direct binding to the *cat-1* promoter in vitro using electrophoretic mobility shift assays (EMSA) (*Figure 5—figure supplement 2*).

The HSN minimal CRM for *bas-1* (AADC) (*bas-1prom18, Figure 4C*) contained predicted binding sites for four TFs from the HSN TF collective: ETS, POU, GATA and SPALT TFs, but lacked any

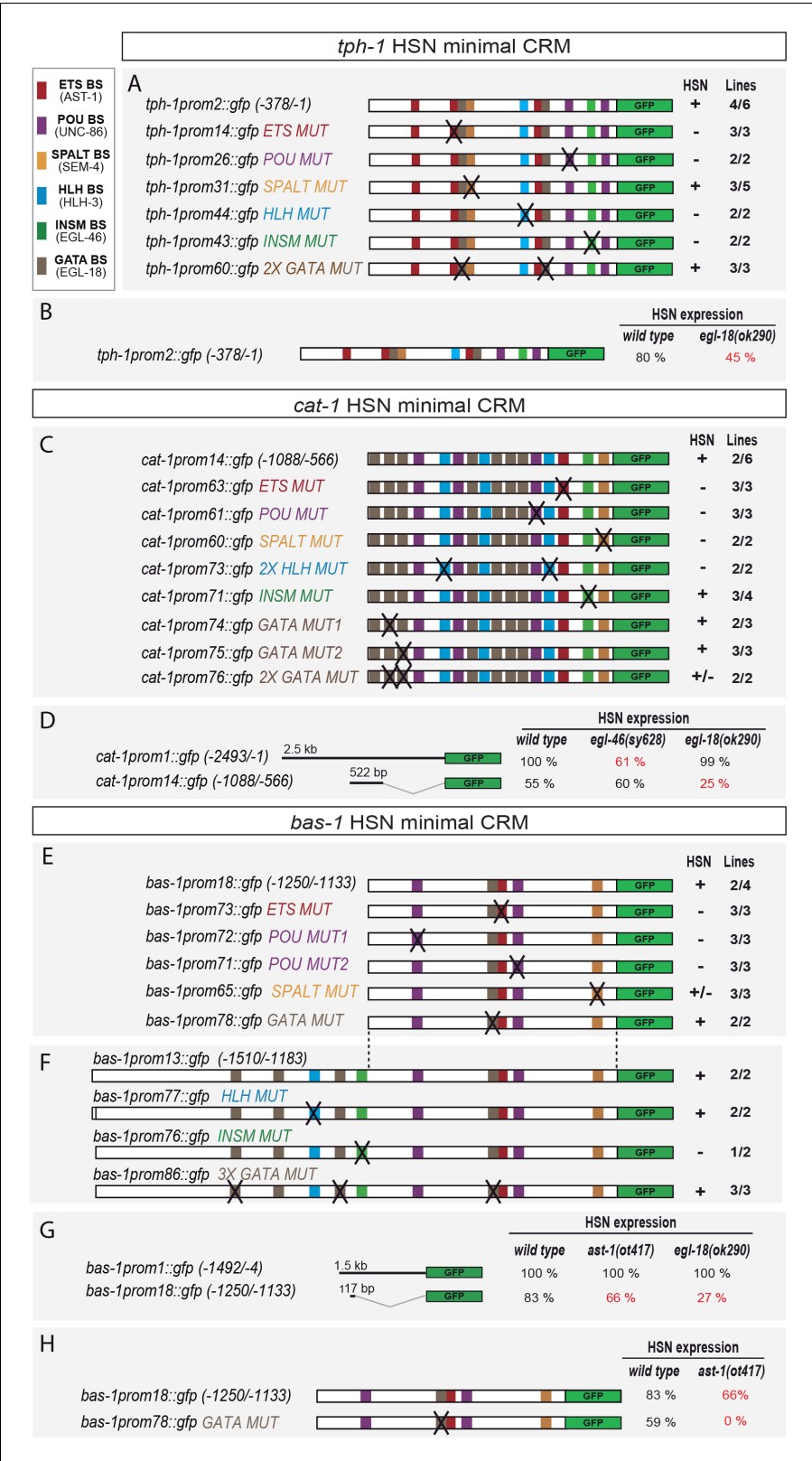

**Figure 5.** HSN transcription factor combination acts directly on target genes. (**A**) *tph-1* minimal HSN CRM (*tph-1prom2*) mutational analysis. Black crosses represent point mutations to disrupt the corresponding TFBS. +: > 60% of mean wild type construct values; +/−: expression values 60–20% lower than mean wild type expression values; −: values are less than 20% of mean wild type values. n > 60 cells per line. x/y represents the number of lines with

*Figure 5 continued on next page*

*Figure 5 continued*

the expression pattern (x) from the total lines analyzed (y). See *Figure 5—figure supplement 1* for raw values and nature of the mutations and *Figure 5—figure supplement 2* for in vitro binding. (B) *tph-1prom2::gfp* expression is partially affected in *egl-18(ok290)* mutants. In red, significant defects relative to wild type. n > 100 cells for each genotype. (C) *cat-1* minimal HSN CRM (*cat-1prom14*) mutational analysis. (D) *cat-1prom14::gfp* expression is unaffected in *egl-46* mutants, which coincides with the lack of phenotype when INSM binding sites are mutated in this construct. *cat-1prom14::gfp* contains functional GATA sites and, as expected, its expression is affected in *egl-18* mutants. Expression of a longer reporter (*cat-1prom1::gfp*) is independent of *egl-18* revealing compensatory effects in the context of big regulatory sequences. (E) *bas-1* minimal HSN CRM (*bas-1prom18*) mutational analysis. (F) A longer *bas-1* construct (*bas-1prom13*) is more robustly expressed in HSN (90% expression compared to mean 48% expression of *bas-1prom18* reporter lines). This construct contains functional INSM binding sites. (G) *bas-1prom18::gfp* expression is affected in *ast-1(ot417)* and *egl-18(ok290)* mutants. Expression of a longer reporter (*bas-1::prom1*) is independent of *ast-1* and *egl-18* revealing compensatory effects in the context of big regulatory sequences. (H) GATA-binding site point mutation does not significantly affect *bas-1::gfp* expression in the wild type background (no significant difference between mean expression of three lines of *bas1prom1* and three lines of *bas1prom18*). However, it synergizes with *ast-1* mutant background leading to a complete loss of GFP expression. These results unravel a direct role for GATA sites in *bas-1* gene expression and synergy between *egl-18* and *ast-1*.

DOI: https://doi.org/10.7554/eLife.32785.012

The following figure supplements are available for figure 5:

**Figure supplement 1.** Primary data from the mutagenesis analysis (*Figure 5*).
DOI: https://doi.org/10.7554/eLife.32785.013

**Figure supplement 2.** UNC-86, EGL-18 and AST-1 bind to the 5HT pathway gene CRMs in electrophoretic mobility assays.
DOI: https://doi.org/10.7554/eLife.32785.014

---

predicted INSM- or HLH-binding sites. Reporter analyses of the minimal CRM revealed that ETS-, POU- and SPALT- but not GATA-binding sites were required for expression in HSN (*Figure 5E* and *Figure 5—figure supplement 1*). Similar to *cat-1*, a *bas-1* functional binding site for several TFs was detectable only in the context of the minimal small CRMs while there was no defect in expression of the full-length reporter in the corresponding TF mutant backgrounds. For example, we found functional ETS- (*ast-1*) binding sites in *bas-1prom18* while expression of the full-length *bas-1* reporter was unaffected in *ast-1 (ot417)* (*Figure 5G*). As *ast-1 (ot417)* is a hypomorphic allele, we confirmed that *ast-1* is not required for *bas-1* full-length reporter expression by mosaic analyses with a rescuing array in a null *ast-1* allele (*hd92*) (87 out of 87 *ast-1* null HSN neurons expressed *bas-1*). We analyzed minimal CRM *bas-1prom18* activity in *ast-1(ot417)* mutants and found a small but significant reduction in the percentage of GFP-positive HSNs (*Figure 5G*). We also confirmed AST-1 binding to the *bas-1* promoter in vitro using EMSA (*Figure 5—figure supplement 2*). Altogether, these results suggest that AST-1 can bind and activate transcription from the *bas-1* minimal CRM as can EGL-18 from *cat-1* minimal CRM. In both cases, however, other factors can compensate for their loss by activating transcription from regulatory sequences outside the minimal CRMs. This genetic redundancy for some members of the HSN TF collective at specific 5HT pathway genes possibly acts as a mechanism to ensure that differentiation is robust.

Although HLH-3 (bHLH) and EGL-46 (INSM) were required for full-length *bas-1* expression (*Figure 1E*), no functional HLH- or INSM-binding sites were found in the minimal *bas-1* CRM (*bas-1prom18*) (*Figure 5E*). Similar to the minimal *cat-1* CRM (*cat-1prom14*), GFP expression of *bas-1prom18* was partially penetrant (ranging from 38% to 83% depending on the transgenic line, *Figure 5—figure supplement 1*), while a longer construct (*bas-1prom13*) was more robustly expressed (90% expression in all lines, *Figure 5—figure supplement 1*). *bas-1prom13* contains bHLH- and INSM-binding sites and INMS-binding site mutation, but not bHLH mutation, leads to expression defects which suggest a direct role for EGL-46 in robust *bas-1* expression (*Figure 5F*).

We did not find functional GATA-binding sites in *bas-1* CRMs, and *egl-18* (GATA) mutants did not show *bas-1* expression defects either. This would suggest that GATA factors are dispensable for the regulation of this gene. However, as we had already observed genetic redundancy in other CRMs, we considered that this could also be the case for *bas-1* regulation. First, we analyzed *bas-1* minimal CRM expression (*bas-1prom18*) in *egl-18(ok290)* mutants and found that EGL-18 was

required for its normal expression (*Figure 5G*). Next, to determine whether the role for GATA factors in *bas-1* expression was direct, we analyzed the expression of a *bas-1* minimal CRM carrying GATA-binding site mutations (*bas-1prom78*) in the *ast-1(ot417)* genetic background. Interestingly, while GATA-binding site mutations had no significant effects in wild type worms, we found a complete loss of expression of this construct in *ast-1(ot417)* mutants (*Figure 5H*). These results revealed both a direct role for GATA factors in *bas-1* expression and redundancy/compensatory effects between *egl-18* and *ast-1*. Interestingly, these two factors do not act redundantly in other CRMs such as *tph-1*.

Of note, despite the fact that HLH-3 expression is not maintained during adulthood (*Figure 2B*) we find functional bHLH-binding sites both in *tph-1* and in *cat-1* CRMs. These results suggest that HLH-3 is directly required to initiate expression of some HSN effector genes. Similar direct action on effector genes has been described for mouse ortholog ASCL1 in the regulation of neuronal differentiation (*Raposo et al., 2015*).

## HSN TF collective shows enhancer-context dependent synergistic relationships

Our *cis*-regulatory analysis revealed compensatory effects among the HSN TF collective, thus, to increase our understanding of the TF collective action, we performed double mutant analysis. We analyzed *tph-1, cat-1* and *bas-1* reporters because their HSN CRMs contain functionally verified binding sites for all six factors (*Figure 5*). *unc-86* and *sem-4* null mutants show complete loss of expression of *tph-1, cat-1* and *bas-1*, thus we used hypomorphic alleles with partial phenotypes for double mutant analysis.

Synergism was the most common effect in our double mutant analysis, although we also found epistatic effects, additivity and suppression (*Figure 6*, *Figure 6—figure supplement 1* and *Figure 6—source data 1*). We found synergism among different members of the HSN TF collective in their action upon *tph-1, cat-1* and *bas-1* reporters (*Figure 6A–H*). Interestingly, the same pair of TFs acting synergistically in the regulation of one reporter can show a different genetic relationship in the regulation of a different gene (*Figure 6E–H*). For example, while *unc-86* acts synergistically with *sem-4* and *hlh-3* in the regulation of *cat-1* expression, it shows additive effects with both TFs in the regulation of *tph-1* reporter (*Figure 6E,F*). Similarly, *hlh-3* shows synergy with *egl-18* and *egl-46* in the regulation of *cat-1* and *tph-1* respectively, while it is epistatic to *egl-18* in the regulation of *bas-1* (*Figure 6G,H*). Reporter specific synergistic effects have been previously described (*Doitsidou et al., 2013*; *Zhang et al., 2014*) and are likely a direct consequence of the flexibility of the TF collective mode of action that shows different disposition of functional binding sites in each enhancer. Unexpectedly, *hlh-3* mutation suppresses *egl-46* phenotype in the regulation of *cat-1* expression (*Figure 6H*). Genetic suppression is an intriguing phenotype that could reflect complex effects of competition for protein-protein interactions. We found additional examples of suppression in our double mutant analysis (*Figure 6—figure supplement 1*)

The HSN TF collective has pleiotropic functions. To try to avoid pleiotropic effects, we took advantage of our *cis*-regulatory data to perform combinations of TFBS mutations. To check for TFBS interactions, our analysis was limited to those sites which mutations produced only partial defects (*Figure 5*). We tested three out of the four possible combined TFBS mutations; however, none of these constructs showed synergistic effects (*Figure 6—figure supplement 1*). Interestingly, double *cis* SPALT- and GATA- BS mutations do not show synergy in the context of the minimal *tph-1* CRM despite the synergistic effect observed for *sem-4, egl-18* double mutants in the regulation of *tph-1* full length reporter. As these two factors do not regulate each other's expression (*Figure 3B*), it is possible that TFBS mutations are more easily compensated than mutations in the corresponding trans-activating factors.

As we were limited by the lack of additional BS mutations with partial effects, we next combined *cis* mutations of the TFBS with *trans* effects of TF single mutants. Combined *cis/trans* mutant analysis revealed synergistic relationships among two additional pairs of the HSN TF collective (*Figure 6I*). Altogether, we found synergistic relationships among 9 out of the 15 possible HSN TF collective pair combinations (*Figure 6—figure supplement 1* and *Figure 6—source data 1*). In addition to synergism, additivity and epistasis, we found several examples of genetic suppression both in the double mutant and the *cis/trans* mutant analysis (*Figure 6I* and *Figure 6—figure supplement 1*). Similar to the other genetic interactions, suppression is also TF pair and enhancer context specific.

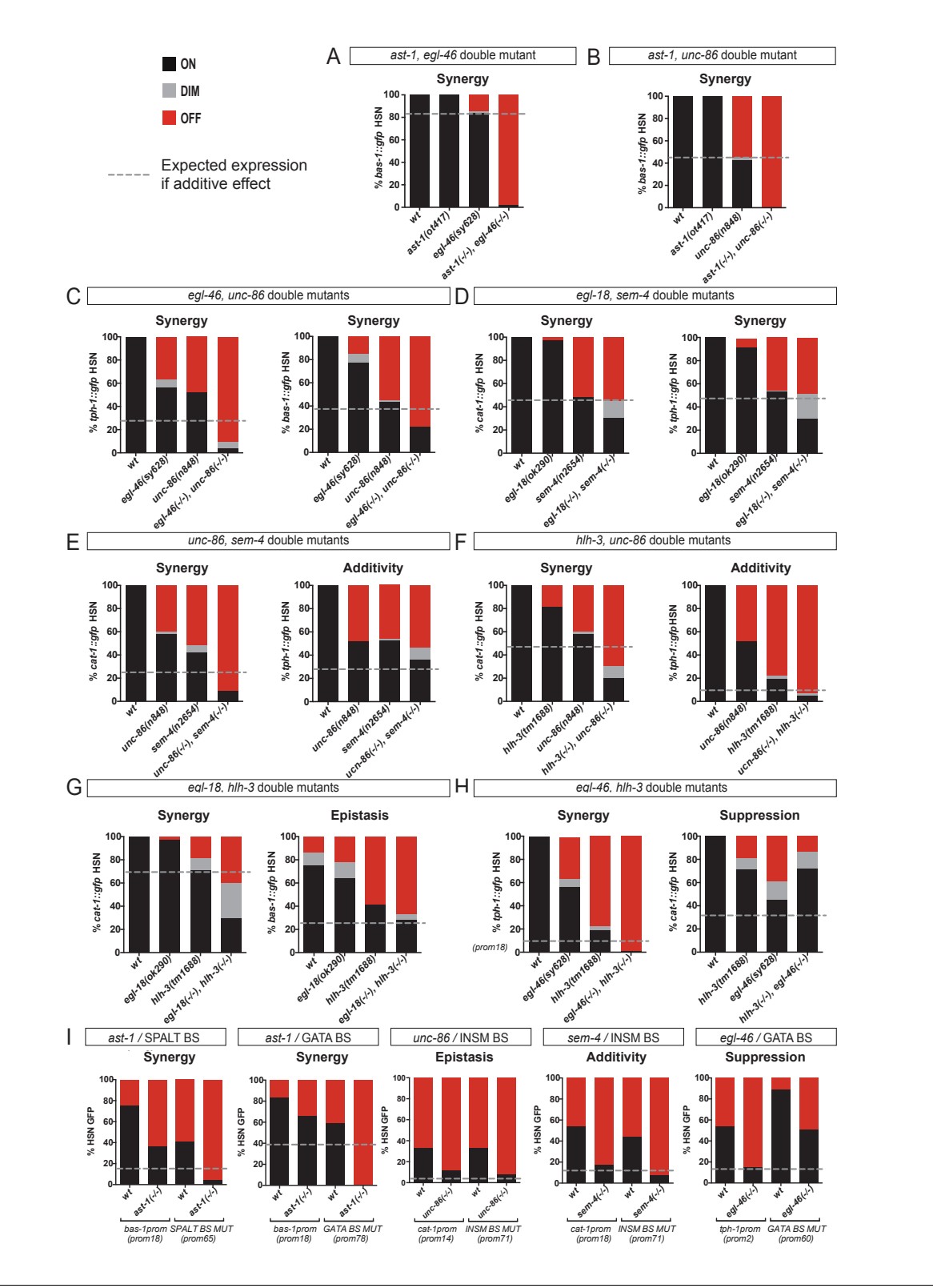

**Figure 6.** HSN TF collective shows enhancer-context dependent synergistic relationships. (A–H) Double mutant analysis of different pairs of the HSN TF collective. Expression level expected from additive effects (calculated as the product of single mutant expression values) is marked with a dotted line. Double mutant phenotypes statistically different from additive effect (Pearson's chi-squared test) are classified as synergistic (if phenotype is stronger than additive), epistatic (if phenotype is similar to one of the single mutants) or suppression (if phenotype is milder than the expected for

*Figure 6 continued on next page*

*Figure 6 continued*

additivity or the single mutants). The majority of the double mutant combinations show synergistic effects. n > 100 cells each genotype. See *Figure 6—figure supplement 1* and *Figure 6—source data 1* for raw values, statistics and additional double mutant combinations. (I) *Cis-trans* mutant analysis. TFBS mutations are combined with single mutants of the HSN TF collective. n > 100 cells each genotype. See *Figure 6—source data 1* raw values and statistics.

DOI: https://doi.org/10.7554/eLife.32785.015

The following source data and figure supplement are available for figure 6:

**Source data 1.** Raw scoring data and statistical analysis of double mutant and double *cis/trans* analysis.

DOI: https://doi.org/10.7554/eLife.32785.017

**Figure supplement 1.** HSN TF collective genetic interactions.

DOI: https://doi.org/10.7554/eLife.32785.016

Altogether, the emerging picture is that of a joint action of the HSN TF collective upon their direct target genes. This regulation is flexible and often partially redundant showing synergistic relationships among different members of the HSN TF collective. Importantly, specific relationships and dependencies are determined by the CRM context and by the specific TF pairs tested.

## The HSN regulatory signature allows de novo identification of HSN expressed genes

Our results suggest that the HSN TF collective is required for broad activation of HSN effector genes (and not only for 5HT pathway gene expression) (*Figure 1*) and it acts directly on the regulatory regions of their target genes (*Figure 5*). Since the members of the HSN TF collective belong to six different TF families that recognize very different binding sites (*Figure 7A*), we wondered whether the clustering of binding sites for the HSN TF collective in regulatory regions of HSN effector genes might confer sufficient specificity to impose a defining regulatory signature.

There are 96 genes known to be expressed in the HSN neuron (*Supplementary file 2*) (*Hobert et al., 2016*), excluding pan-neuronal features which are regulated in a very redundant manner (*Stefanakis et al., 2015*). We analyzed upstream and intronic sequences of HSN expressed genes in search of DNA windows (up to 700 bp length) containing at least one position weight matrix match for all six members of the HSN TF collective (termed the 'HSN regulatory signature') (*Figure 7A*). We found that known HSN expressed genes contained large upstream and intronic sequences, thus, for comparison purposes, we built ten thousand sets of 96 random genes with similar upstream and intronic length distribution. A significantly higher percentage of HSN expressed genes contain the HSN regulatory signature compared to the random sets of genes (p<0.05) (*Figure 7B*, *Figure 7—source data 1*).

Studies in *Drosophila* and vertebrates have shown that functional enhancers that are bound by combinations of TFs show higher interspecific conservation compared to enhancers bound by single TFs (*Ballester et al., 2014*; *Khoueiry et al., 2017*; *Stefflova et al., 2013*). Thus, we performed a similar motif search in *C. brenneri*, *C. remanei*, *C. briggsae* and *C. japonica* genomes and calculated, for each *C. elegans* gene, the proportion of its orthologs that had, in its upstream or intronic sequence, at least one 700 bp window with binding sites for all the six TFs. We considered the HSN regulatory signature as phylogenetically conserved when orthologous genes in all species displayed the signature within their upstream or intronic regions. We found that the inclusion of the conservation criteria in this analysis slightly increased the difference between HSN and the random sets of genes (p<0.01) (*Figure 7B*).

The higher prevalence of conserved signature in HSN expressed genes supports the idea that the HSN TF collective broadly selects the HSN transcriptome. We tested HSN regulatory signature windows from four of the known HSN expressed genes by in vivo reporter assays and confirmed that they correspond to active HSN enhancers (three out of four tested contructs show HSN expression, *Figure 7—figure supplement 1* and *Figure 7—source data 2*). Of note, *C. elegans* functional HSN regulatory signature windows do not show a high level of sequence conservation (*Figure 7—figure supplement 1*), which is in agreement with rapid evolution of regulatory sequences (*Villar et al., 2015*).

Next, we examined the distribution of the HSN regulatory signature windows across the entire *C. elegans* genome. Remarkably, we found that it was preferentially found in the putative regulatory

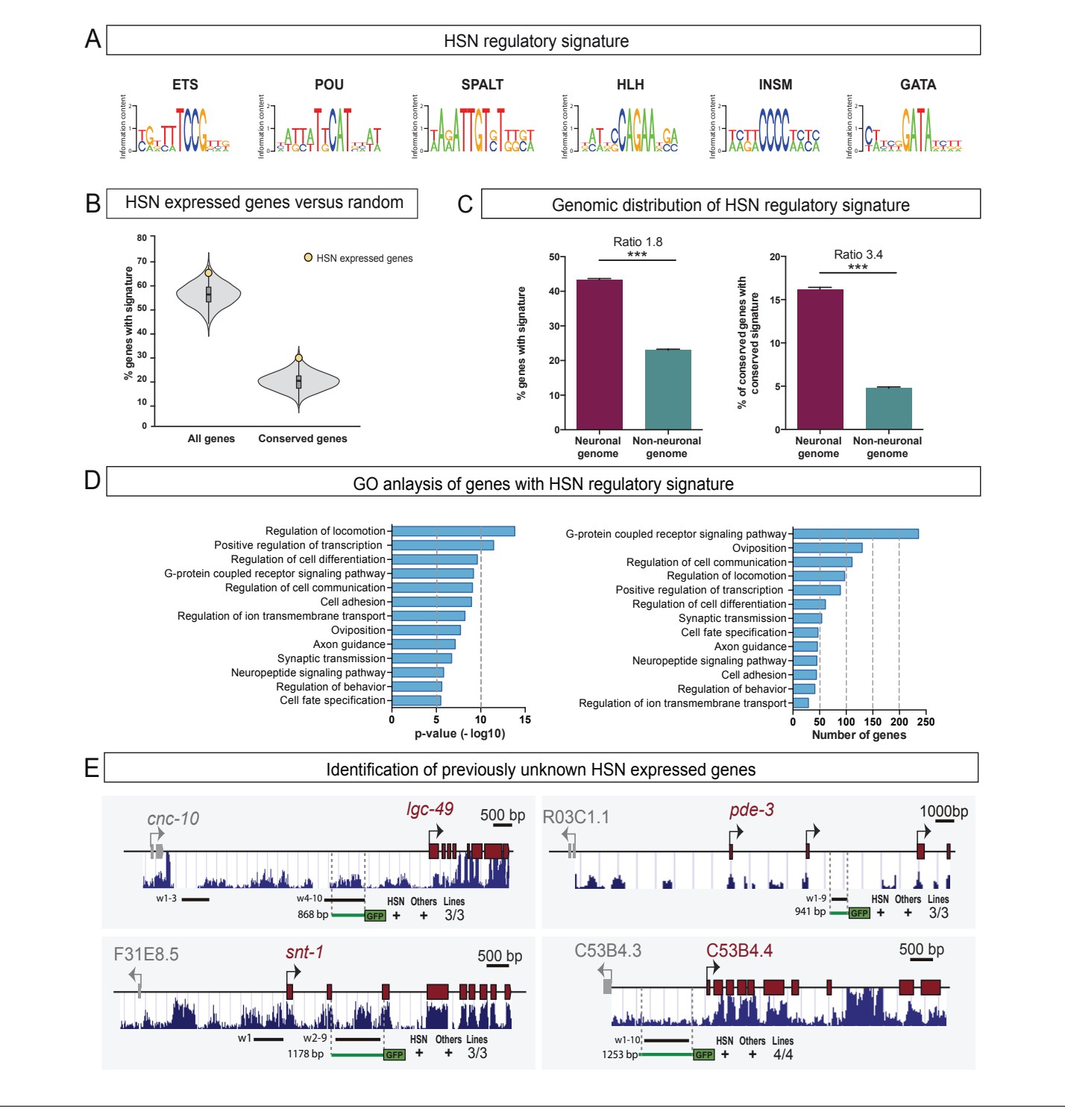

**Figure 7.** The HSN regulatory signature can be used to de novo identify HSN expressed genes. (**A**) Position weight matrix logos of the HSN TF collective calculated from the functional binding sites in **Figure 5**. (**B**) HSN regulatory signature is more prevalent in the set of 96 known HSN expressed genes (yellow dot) compared to the distribution in 10,000 sets of random comparable genes (grey violin plot) (p<0.05). Considering phylogenetic conservation of HSN regulatory signature increases the enrichment of the HSN regulatory signature in the HSN expressed genes (p<0.01). See also **Figure 7—figure supplement 1** for additional data. (**C**) HSN regulatory signature is enriched in neuronal genes compared to the non-neuronal genome. Inclusion of the conservation criteria in the HSN regulatory signature analysis strongly increases the difference between neuronal and non-neuronal genome. Pearson's chi-squared test. ***p-value<0.0001. See also **Figure 7—figure supplement 2** for additional data. (**D**) Gene ontology analysis of genes with HSN regulatory signature. p-values and number of genes corresponding to the biological processes enriched in genes with HSN

*Figure 7 continued on next page*

*Figure 7 continued*

regulatory signature. (E) Four representative examples of de novo identified HSN active enhancers. Black lines represent the coordinates covered by bioinformatically predicted HSN regulatory signature windows (indicated by 'w' and a number). Green lines mark the region used in our analysis. Dark blue bar profiles represent sequence conservation in *C. briggsae, C. brenneri, C. remanei* and *C. japonica.* n > 60 cells per line. See *Figure 7—source data 1* for a list of all reporters and raw scoring data. Expression level of most of these reporters is regulated by *unc-86* (*Figure 7—figure supplement 3*).

DOI: https://doi.org/10.7554/eLife.32785.018

The following source data and figure supplements are available for figure 7:

**Source data 1.** Scripts for HSN regulatory signature analysis.
DOI: https://doi.org/10.7554/eLife.32785.022
**Source data 2.** Raw scoring data of de novo finding of HSN enhancers and dependency on *unc-86* function.
DOI: https://doi.org/10.7554/eLife.32785.023
**Figure supplement 1.** HSN regulatory signature distribution in HSN expressed genes.
DOI: https://doi.org/10.7554/eLife.32785.019
**Figure supplement 2.** Analysis of the HSN regulatory signature including windows missing one or two TFBS motifs.
DOI: https://doi.org/10.7554/eLife.32785.020
**Figure supplement 3.** Expression of identified HSN regulatory windows depends on *unc-86*.
DOI: https://doi.org/10.7554/eLife.32785.021

sequences of genes known to be expressed in neurons or that have a neuronal function compared to the rest of the genome, as would be expected for genes controlled by the HSN TF collective (*Figure 7C*). Filtering of conserved regulatory signatures further increased the difference between 'neuronal' and 'non-neuronal' genomes, which adds support to its functionality (*Figure 7C*). Gene ontology analysis of all genes in the *C. elegans* genome with HSN regulatory signature revealed enrichment of processes controlling transcription, axon guidance, synaptic transmission and oviposition, all characteristic of HSN differentiation and function (*Figure 7D*).

Our experimental data (*Figure 5*), in agreement to the TF collective model (*Spitz and Furlong, 2012*), shows that the presence of TFBS for all TF collective members is not required in specific enhancer contexts. Thus, we aimed to analyze if HSN regulatory windows lacking TF-binding sites for one or two TF classes show also an enriched distribution in HSN expressed genes and in the neuronal genome. We find that, in contrast to the six-motif HSN regulatory signature, windows containing only five or four types of HSN TF motifs are not preferentially found in HSN expressed genes compared to the 10.000 random sets of genes with or without filtering for conservation (p>0.05 in all conditions) (*Figure 7—figure supplement 2*). Additionally, genomic distribution of the HSN regulatory signature is less enriched in neuronal genes compared to non-neuronal genes when including windows lacking one or two HSN TF collective motifs (*Figure 7—figure supplement 2*). Moreover, while only 25% of the genes (4,968) contain at least one assigned six-motif regulatory window, regulatory windows with five or more motifs are found in 52% of the genes (10,415) and 72% of the genes (14,325) contain windows with four or more motifs. Finally, GO comparative analysis shows that genes with assigned 6-motif HSN regulatory windows show the highest enrichment in terms related to HSN function and that the additional GO terms obtained when including windows lacking either one or two HSN TF motifs are not related to neuronal functions (*Figure 7—figure supplement 2*). Altogether, our data shows that the most prevalent mark of HSN expressed genes is the regulatory signature with all six TFBS. Even if some HSN enhancers can still be functional with a partial complement of HSN TF collective binding sites, at the genomic level, including enhancers with missing TFBS abolishes cell type specificity.

Next, we aimed to identify new genes expressed in HSN based solely on the presence of the HSN regulatory signature. To this end, we randomly selected 35 neuronal genes with a conserved HSN regulatory signature and generated transgenic reporter lines. We found that 13 out of the 35 constructs (37%) showed GFP expression in HSN (*Figure 7E* and *Figure 7—source data 2*), while none of 10 randomly picked similar-sized intergenic regions of neuronal genes lacking the HSN regulatory signature led to reporter expression in HSN (*Figure 7—source data 2*). Importantly, all reporter constructs, including the negative controls, did drive GFP expression in a variable set of additional neurons, which might be due to the compact nature of the *C. elegans* genome.

Finally, to analyze if the activity of the identified HSN regulatory windows was under the control of the HSN TF collective we crossed them into the *unc-86(n846)* mutant. The expression of 12 out of 15 reporter constructs (80%) was significantly reduced in *unc-86* mutants (*Figure 7—figure supplement 3*). Of note, onset of expression of the HSN regulatory window reporters can be used to predict the effect of *unc-86* mutation: while all reporters with L4 onset of expression are strongly dependent on *unc-86*, HSN regulatory windows that initiate expression at earlier stages show more modest dependency on *unc-86* function (*Figure 7—figure supplement 3*).

Our results reveal that the presence of a conserved HSN regulatory signature can be successfully used to de novo identify HSN expressed genes. However, our high level of false positives (63%) indicates that the signature itself is not sufficient to induce HSN expression. Additional TFs might be part of the HSN TF collective and thus active HSN regulatory signature windows would contain additional TFBS. Repressive elements or chromatin accessibility could also block HSN expression of non-functional HSN regulatory signature windows, indeed members of the SWI/SNF chromatin remodeling complex are required for correct HSN terminal differentiation (*Weinberg et al., 2013*). It is also possible that specific syntactic rules (TFBS order, distance and disposition) discriminate functional from non-functional HSN regulatory signature windows. Future studies will help identify additional players and rules for HSN terminal differentiation.

## Deep homology between HSN and mouse raphe serotonergic neurons

Mouse orthologs for four out of the six TFs of the HSN TF collective are involved in mammalian serotonergic specification: ASCL1 (bHLH TF ortholog of HLH-3) (*Pattyn et al., 2004*), GATA2/3 (GATA TF ortholog of EGL-18) (*Haugas et al., 2016*), INSM1 (Zn Finger Insm TF ortholog of EGL-46) (*Jacob et al., 2009*) and PET1 (ETS TF ortholog of AST-1) (*Hendricks et al., 2003*). Additionally, BRN2 (also known as POU3F2, a POU TF from the same family that UNC-86) has been associated with serotonergic specification, although its expression in serotonergic neurons has not been studied (*Nasu et al., 2014*). We analyzed BRN2 expression in serotonergic differentiating neurons and found it expressed in serotonergic progenitors and serotonergic newborn neurons at embryonic stage E11.5, when serotonergic neurons are differentiating (*Figure 8A and B*). Finally, SALL2 is the closest mouse ortholog for *C. elegans* SEM-4, but there is no known role for any SALL TFs in serotonergic specification. We found that SALL2 is also expressed in serotonergic progenitors and serotonergic newborn neurons at embryonic stage E11.5 (*Figure 8A and C*).

In evolutionary biology, the term deep homology refers to the relationship between two structures that share the genetic mechanisms governing their differentiation (*Shubin et al., 1997*). As *C. elegans* HSN and mouse raphe serotonergic neurons share many of the TFs required for their differentiation, we hypothesized that they might be homologous structures. If this were the case, then HSN neurons and mouse serotonergic raphe should not merely share the expression of 5HT pathway genes, which are also present in the other *C. elegans* serotonergic neuron classes, but also be more broadly similar in molecular terms.

To address this, we used available gene expression data from the WormBase to generate partial expression profiles for the 118 neuronal classes of the *C. elegans* hermaphrodite. This partial expression profile can be successfully used to reproduce the anatomical classification of *C. elegans* neuron subtypes (*Hobert et al., 2016*). We assigned mouse orthologs to *C. elegans* neuronal genes and merged the resulting table with another one featuring the available mouse raphe serotonergic neuron transcriptome (*Okaty et al., 2015*). Hierarchical clustering of this data set shows that HSN is, molecularly, the closest neuron to the mouse raphe neurons (*Figure 8D*). Importantly, hierarchical clustering generated from mouse orthologs of *C. elegans* genes resembles the neuron class clustering generated directly from *C. elegans* genes (*Hobert et al., 2016*). HSN and mouse raphe close relationship is not merely due to 5HT pathway genes expression because NSM and ADF serotonergic neurons are molecularly more distant to the mouse raphe serotonergic neurons than HSN (*Figure 8D* and Source Data 5). Moreover, HSN remained the most similar neuron to mouse serotonergic raphe neurons even after removing the 5HT pathway genes from the HSN expression profile (*Figure 8—figure supplement 1*). Shared orthologous genes between HSN and mouse raphe serotonergic neurons belong to different functional categories including axon guidance and migration, neurotransmission, or transcriptional regulation (*Table 1*). Importantly, HSN proximity to mouse serotonergic neurons was not maintained with other mouse neuronal populations (*Figure 8—figure supplement 1*).

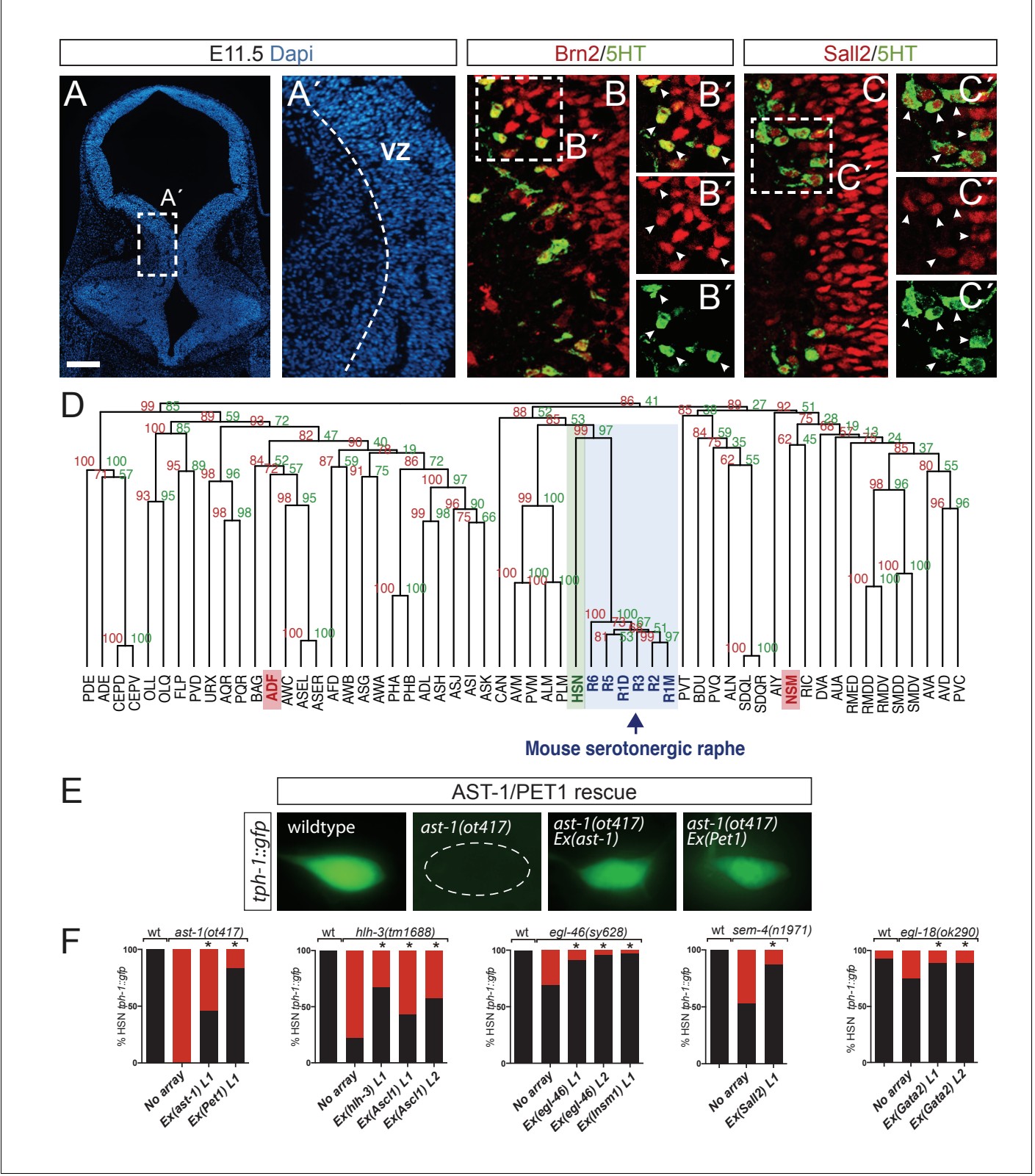

**Figure 8.** Deep homology between HSN and mouse serotonergic raphe neurons. (**A**) Micrograph of mouse embryonic day 11.5 hindbrain coronal section with DAPI staining. Square box indicates the region in A', B and C panels. VZ: ventricular zone, where progenitors are located. Scale bar: 100 µm. (**B**) BRN2 and 5HT co-staining. BRN2 is expressed in progenitors and differentiating serotonergic neurons. Arrowheads indicate double labeled cells. Scale bar: 20 µm. (**C**) SALL2 and 5HT co-staining. SALL2 is expressed in progenitors and differentiating serotonergic neurons. Arrowheads indicate

*Figure 8 continued on next page*

*Figure 8 continued*

double labeled cells. Scale bar: 20 µm. (D) Hierarchical clustering analysis of *C. elegans* neuron expression profiles with mouse serotonergic raphe neurons shows that HSN (in green) is closest to mouse serotonergic neurons (in blue). Other *C. elegans* serotonergic neuron classes (ADF and NSM in red) do not show a close relationship with mouse serotonergic raphe. R1D: Dorsal serotonergic neurons from rhombomere r1; R1M: Medial serotonergic neurons from rhombomere r1; R2: serotonergic neurons from rhombomere r2; R3: serotonergic neurons from rhombomere r3; R5: serotonergic neurons from rhombomere r5; R6: serotonergic neurons from rhombomere r6. See also *Figure 8—figure supplement 1*. (E) Micrographs showing *tph-1::gfp* expression in wild type animals, *ast-1(ot417)* mutants, and *ast-1(ot417)* mutants rescued with *ast-1* cDNA or mouse Pet1 cDNA expressed under the *bas-1* promoter whose expression in not affected in this mutant background. (F) Quantification of *tph-1::gfp* HSN expression rescue of different HSN TF collective mutants with worm and mouse ortholog cDNAs. n > 100 cells per condition. Fisher's exact test, *: p-value<0.05. 'L' indicates the transgenic line number.

DOI: https://doi.org/10.7554/eLife.32785.024

The following source data and figure supplement are available for figure 8:

**Source data 1.** Scripts for *C.elegans* and mouse neuron comparison.
DOI: https://doi.org/10.7554/eLife.32785.026
**Figure supplement 1.** HSN neuron is the *C. elegans* neuron molecularly closest to mouse raphe serotonergic neurons.
DOI: https://doi.org/10.7554/eLife.32785.025

Finally, to test if there is deep homology between HSN and mouse raphe serotonergic neurons, we tested if mouse orthologs of the HSN TF collective can functionally substitute for their worm counterparts. We performed cell-specific rescue experiments of *C. elegans* mutants and found that mouse Pet1, Ascl1, Insm1, Gata2 and Sall2 could respectively substitute *ast-1*, *hlh-3*, *egl-46*, *egl-18* and *sem-4*, which suggest that this regulatory program could be phylogenetically conserved (*Figure 8E and F*). Of note, our rescue experiments, both with *C. elegans* or mouse genes, restore *tph-1* expression but do not rescue egg-laying defects. The HSN TF collective has pleiotropic actions in other tissues that also contribute to the egg laying phenotype (*Basson and Horvitz, 1996*; *Doonan et al., 2008*; *Eisenmann and Kim, 2000*; *Koh et al., 2002*) what could explain the persistence of egg-laying defects in the HSN specific rescue experiments.

In sum, these results revealed an unexpected level of regulatory and molecular proximity between *C. elegans* HSN and mouse serotonergic raphe neurons suggesting that deep homology might exist between these two neuronal types.

## Discussion

Our extensive analysis of the HSN regulatory logic has revealed insights into how the complement of cell-type-specific enhancers is selected. We found that numerous TFs (here we identify six but likely additional TFs are required) act in conjunction to directly activate the HSN regulatory landscape. HSN TF collective acts through the HSN regulatory signature, which is found preferentially associated to genes of the neuronal genome that are related to HSN function.

### Regulation of *C. elegans* neuron specification is unexpectedly complex

Neuronal terminal differentiation programs have been best characterized in *C. elegans*. So far, relatively simple TF combinations, composed of two or three members, were shown to be required, and in some contexts sufficient, to select specific neuronal types. These TFs have been termed Terminal Selectors (*Doitsidou et al., 2013*; *Serrano-Saiz et al., 2013*; *Van Buskirk and Sternberg, 2010*; *Zhang et al., 2002*). In some cases, additional TFs act together with Terminal Selectors to partially modulate the transcriptomes of specific neuronal subclasses (*Kerk et al., 2017*; *Kratsios et al., 2017*). Accordingly, it has been suggested that, in *C. elegans*, a rather simple organization of CRMs control the expression of neuronal terminal features (*Holmberg and Perlmann, 2012*). Our results, however, demonstrate a more complex scenario in the regulation of the HSN transcriptome. We have identified six TFs required for HSN terminal differentiation acting directly upon the regulatory regions of HSN expressed genes. Nonetheless, additional unidentified factors are likely to compose the HSN TF collective. We found that the HSN TF collective includes a proneural TF (*hlh-3*) that is required to initiate HSN differentiation but whose expression, like all proneural factors (*Guillemot and Hassan, 2017*), is not maintained in the mature neuron. Future experiments should determine if, as has been proven for its mouse ortholog Ascl1 (*Wapinski et al., 2013*), *hlh-3* acts as

**Table 1.** *C. elegans* HSN and mouse raphe serotonergic neuron homology

| *C. elegans* gene name | Description | Mammalian gene name |
|---|---|---|
| **Serotonergic biosynthetic pathway** | | |
| *bas-1* [a] | Dopamine decarboxylase | Ddc |
| *cat-1*[a] | Vesicular monoamine transporter | Slc18a2 |
| *cat-4* | GTP cyclohydrolase 1 | Gch1 |
| *tph-1* [a] | Tryptophan hydroxylase | Tph2 |
| **Axon guidance and Migration** | | |
| *ebax-1* [a] | Elongin-B/C E3 ligase | Zswim5/6/8 |
| *egl-43* [a] | PR domain containing | Prdm16 |
| *fmi-1* | Flamingo homolog | Celsr2/3, Fat1/3, Dchs1 |
| *madd-2* [a] | Trim protein | Trim9/36/46, Fsd1/1 l, Mid2 |
| *mau-2* [a] | Chromatid cohesion factor | Mau2 |
| *mig-10* [a] | Protein with an RA-like, PH domains and proline-rich motif | Raph1, Grb10 |
| *nck-1* | SH2/SH3 domain-containing protein | Nck1 |
| *rig-6* [a] | neuronal IgCAM | Cntn1, 2, 3, 4, 5, 6 |
| *tbb-4* [a] | Tubulin | Tubb2a/2b/4a/4b/5 |
| *unc-40* [a] | Netrin receptor | Dcc, Neo1 |
| *unc-51* [a] | Serine/threonine protein kinase | Ulk1/2 |
| *unc-53* [a] | Neuron navigator | Nav1/2/3 |
| **Neurotransmission/Synaptogenesis** | | |
| *abts-1* [a] | Anion/Bicarbonate Transporter family | Slc4a7/8/10 |
| *clh-3* | Voltage sensitive cloride channel | Clcn2 |
| *eat-16* | Regulator of G protein signaling | Rgs11/19 |
| *gar-2* [a] | G-protein-coupled acetylcholine receptor | Hrh3 |
| *ggr-2* [a] | GABA/Glycine Receptor | Glra1/2, Glrb |
| *glr-5* [a] | Glu Receptor | Grid1/2, Grik1 |
| *gsa-1* [a] | G protein, Subunit Alpha | Gnal, Gnas |
| *ida-1* [a] | Protein tyrosine phosphatase-like receptor | Ptprn, Ptpm2 |
| *irk-1* [a] | Inward Rectifying K (potassium) channel family | Kcnj3/5/6/9/11/16/ |
| *kcc-2* [a] | K/Cl cotransporter | Slc12a5/6 |
| *mpz-1* [a] | Multiple PDZ domain protein | Mpdz, Pdzd2, Inadl, Lnx1 |
| *nhx-5* | Na/H exchanger | Slc9a6/7/9 |
| *nid-1* [a] | Nidogen (basement membrane protein) | Lrp1/1b |
| *nra-4* | Nicotinic Receptor Associated | Nomo1 |
| *rsy-1* | Regulator of synapse formation | Pnisr |
| *syg-1* [a] | Ig transmembrane protein | Kirrel, Kirrel3 |
| *nlg-1* [a] | Neuroligin family | Nlg1/2/3 |
| *unc-2* [a] | Calcium channel alpha subunit | Cacna1a/1b/1e |
| *unc-77* | Voltage-insensitive cation leak channel | Nalcn |
| *unc-103* [a] | K + channel | Kcnh2/7 |
| **Transcriptional regulation** | | |
| *ceh-20* [a] | PBX TF | Pbx1/2/3 |
| *egl-44* [a] | TEA domain TF | Tead1 |
| *gei-8* | Nuclear receptor co-repressor | Ncor1 |
| *hlh-3* [a] | bHLH TF | Ascl1 |

*Table 1 continued on next page*

Table 1 continued

| C. elegans gene name | Description | Mammalian gene name |
| --- | --- | --- |
| ife-4 [a] | Initiation factor 4E | Eif4e2 |
| sem-4 [a] | Spalt TF | Sall2,Zfp236/Znf236 |
| Morphogenetic pathways | | |
| dsh-1 [a] | Homolog of disheveled | Dvl1/3 |
| plr-1 [a] | Ring finger protein | Rnf215 |
| prkl-1 [a] | Drosophila Prickle homolog | Prickle1/2 |
| sel-10 [a] | Suppressor/Enhancer of Lin-12(Notch) | Fbxw7 |
| Others | | |
| aak-2 [a] | AMP-activated protein kinases | Prkaa1/2 |
| ags-3 | G protein singalling modulator | Gpsm1 |
| aho-3 [a] | Hydrolase | Abhd17a/17b |
| ari-1 | Ubiquitin-protein transferase | Arih1 |
| arr-1 | G protein singaling adaptor | Arrb1/2 |
| arrd-17 [a] | Arrestin domain protein | Arrdc3 |
| baz-2 | Bromodomain adjacent to zinc finger domain | Baz2a/2b |
| elpc-1 | Elongator complex protein component | Ikbkap |
| elpc-3 | Elongator complex protein component | Elp3 |
| goa-1 [a] | G protein,O, Alpha subunit | Gnao1 |
| kin-20 | Protein kinase | Csnk1d/1e |
| puf-9 | Pumilio/FBF domain-containing | Pum1/2 |
| pxf-1 [a] | Rap guanine nucleotide exchange factor | Rapgef2/6 |
| rep-1 | Rab escort protein | Chm, Chml |
| ten-1 [a] | Type II transmembrane EGF-like repeats | Tenm1/3/4 |
| top-1 [a] | Topoisomerase | Top1/1mt |

[a] : gene with assigned HSN regulatory signature

DOI: https://doi.org/10.7554/eLife.32785.027

a pioneer factor for HSN terminal differentiation. In light of our findings, nematode neuronal terminal differentiation programs are not necessarily simpler than those found in vertebrates, as previously proposed (**Holmberg and Perlmann, 2012**).

Considering the technical advantages of *C. elegans* as a simple model system, our work is an example on how its study may help to identify rules of terminal differentiation in eumetazoa. The combination of our extensive *cis*-regulatory analysis and the double mutant characterization allowed us to describe the flexible action of the HSN TF collective that can activate enhancers with very different dispositions of TFBS. This flexibility is also made evident by the specific synergistic relationships in the regulation of some enhancers and not others. We propose that these redundant actions, globally considered, confer robustness of expression to the system.

## The HSN regulatory signature identifies HSN expressed genes

Co-binding of specific combinations of TFs to the same genomic region, assessed by ChIP-seq, has been successfully used to identify, de novo, cell-type-specific enhancers in *Drosophila* embryos (**Busser et al., 2015**; **Junion et al., 2012**; **Zinzen et al., 2009**). However, this approach fails to address why specific genomic regions work as enhancers. Recently, massively parallel reporter assays (MPRA) have been used to identify generic rules of enhancer function. The analysis of synthetic enhancers revealed that highest levels of expression are achieved with clusters of binding sites for

different TFs (*Smith et al., 2013*). Another MPRA study has analyzed enhancer activity of regions bound by the adipocyte terminal selector PPARγ and has determined that the best predictor for enhancer functionality is the presence of nearby TFBS for more than 30 different TFs expressed in adipocytes (*Grossman et al., 2017*). In accordance to this complex scenario of combinatorial action of multiple TFs in the global selection of cell type regulatory landscapes, we found that clusters of bioinformatically predicted TFBS for the HSN TF collective can be used for the de novo identification of HSN enhancers. Of note, our analysis still shows a high rate of false positives, which suggests that additional features are present in HSN functional enhancers. Future analyses based on more complex paradigms should facilitate the identification of such features that could include additional TFBS (or the absence of repressor sites) or specific syntactic rules.

## Deep homology, molecular homology and functional homology between *C. elegans* HSN and mouse serotonergic neurons

The diversity of *C. elegans* serotonergic neuronal classes (NSM, ADF and HSN) contrasts with that of tetrapod vertebrates, in which serotonergic neurons are limited to the raphe system (*Flames and Hobert, 2011*). Other chordates contain additional serotonergic populations (*Flames and Hobert, 2011*) and serotonergic subclass diversity is also prevalent in other phyla such as arthropoda and mollusca (*Flames and Hobert, 2011*), which suggests a loss of serotonergic diversity in the tetrapod branch. As in nematodes, serotonergic subclass specification in other organisms is likely to be independently regulated. For instance, in *Drosophila*, the TFs *islet, hunchback* and *engrailed* are required for serotonergic specification of the ventral ganglion, while are dispensable for fly brain serotonergic specification (*Lundell et al., 1996*; *Thor and Thomas, 1997*). Similarly, in zebrafish, Pet1 regulates raphe serotonergic specification but is dispensable for the specification of other serotonergic subclasses (*Lillesaar et al., 2007*).

Considering the homologous regulatory network between HSN and mouse raphe and their molecular proximity, our results suggest that the *C. elegans* HSN serotonergic neuron, but not the NSM or ADF, could share deep homology with mouse raphe neurons. It would be interesting to explore if NSM or ADF regulatory programs show homology to any of the programs controlling the non-raphe serotonergic populations present in other organisms. Noteworthy, despite the homology in TFs regulating HSN and raphe specification, both systems also show discrepancies. For example, while the LIM TF Lmx1b is known to be a key player in mouse serotonergic differentiation (*Ding et al., 2003*), we failed to identify a similar role for any *C. elegans* LIM TF (A.JM and N.F unpublished). Similarly, while *C. elegans* GATA factor *egl-18* has very redundant effects on HSN differentiation, GATA2/3 factors are fundamental in mouse serotonergic differentiation (*Haugas et al., 2016*). Considering the evolutionary distance between mammals and nematodes, the complexity of the regulatory network (composed in *C. elegans* at least by six and most likely more factors) and the fast evolutionary rate of regulatory regions, it is conceivable that the ancestral common serotonergic regulatory network has significantly diverged between these two animal groups. We propose that this deep homology might be the result of a common ancestor cell type, although, as we do not have enough information about the serotonergic regulatory programs in other animal groups, an alternative scenario is that they might have arisen independently in nematodes and vertebrates and thus although some components would have been convergently employed others could be species specific.

If HSN and mouse raphe serotonergic neurons were homologous cell types, we would predict that they are also functionally homologous. Serotonergic systems in all animal groups function as facilitators of motor output, with 5HT promoting a switch between states (*Gillette, 2006*). Interestingly, *C. elegans* 5HT signaling in HSN neurons also facilitates motor output. Egg-laying behavior transitions from inactive to active states of egg laying, and 5HT signaling in HSN mediates the onset of the active phase (*Waggoner et al., 1998*). Thus, HSN and mouse serotonergic neurons would share deep homology, as well as molecular and functional homology.

Deep homology of specific nervous system structures has been previously proposed. Conserved TF expression patterns in annelid antero-posterior nervous system axis, including the serotonergic progenitor region, was used to propose the existence of a common Bilaterian ancestor with centralized nervous system (*Denes et al., 2007*; *Tomer et al., 2010*). Additionally, homologous TF expression patterns have also been used to propose the presence of a visceral nervous system in the common Bilaterian ancestor (*Nomaksteinsky et al., 2013*). Altogether, these results suggest that,

despite considerable divergence in neuronal architecture and connectivity, deep homology could underlie the specification of diverse neuron subtypes. The identification of homologous regulatory programs could help identify homologous neuronal types in distant species.

# Materials and methods

## Key resources table

| Reagent type (species) or resource | Designation | Source or reference | Identifiers |
|---|---|---|---|
| Strain, strain background (*Caenorhabditis elegans*) | C. elegans: Strain N2 | Caenorhabditis Genetics Center | WormBase: N2 |
| Strain, strain background (*Caenorhabditis elegans*) | Strain names and genotypes | *Supplementary file 3* | |
| Strain, strain background (*Mus musculus*) | Mouse: C57Bl/6JRccHsd strain | ENVIGO, Harlan. (Huntingdon, Cambridgeshire, UK) | C57Bl/6JRccHsd |
| Strain, strain background (*Escherichia coli*) | Strain OP50 | Caenorhabditis Genetics Center | OP50 |
| Strain, strain background (*Escherichia coli*) | Rosetta 2(DE3) Singles Competent Cells: BL21 derivatives | Novagen, Merck Group. (Darmstadt, Germany) | Cat#71400 |
| Strain, strain background (*Escherichia coli*) | Strain: HT115(DE3) | Caenorhabditis Genetics Center | HT115 |
| Cell line (human) | Human: HEK293T | Laboratory of Oliver Hobert | ATCC: CRL-3216 |
| Transfected construct (*C.elegans*) | Plasmid: pCDNA3-*egl-18* | This paper | N/A |
| Antibody | Mouse anti-GFP IgG1K | Sigma Aldrich, Merck Group. (Darmstadt, Germany) | Cat#11814460001 |
| Antibody | Anti-6X His tag antibody [HIS.H8] | Abcam (Cambridge, UK) | Cat#ab18184 |
| Antibody | Rabbit anti-5HT | Sigma Aldrich | S5545 |
| Antibody | Goat anti-5HT | Abcam | Ab66047 |
| Antibody | Rabbit anti-Sall2 | Sigma Aldrich | sc-6029 |
| Antibody | Alexa 555-conjugated donkey anti-rabbit | Molecular Probes, Invitrogen (Eugene, OR) | A-31572 |
| Antibody | Alexa 555-conjugated donkey anti-goat | Molecular Probes | A-21432 |
| Antibody | Alexa 488-conjugated donkey anti-rabbit | Molecular Probes | A-21206 |
| Antibody | Alexa 488-conjugated donkey anti-goat | Molecular Probes | A-11055 |
| Recombinant DNA reagent | Plasmid: pPD95.75 | Dr Oliver Hobert Laboratory | Addgene Plasmid #1494 |
| Recombinant DNA reagent | Plasmid: pRF4 (*rol-6 (su1006)*) | (*Mello et al., 1991*) | N/A |
| Recombinant DNA reagent | Plasmid: *ttx-3prom*::mcherry | (*Bertrand and Hobert, 2009*) | N/A |
| Recombinant DNA reagent | Plasmid: pBluescript | Dr Oliver Hobert Laboratory | N/A |
| Recombinant DNA reagent | Plasmid pJJR82 | Dr Mike Boxem Laboratory | Addgene #75027 |
| Recombinant DNA reagent | Plasmid pDD162 | Dr Mike Boxem Laboratory | Addgene #4754 |
| Recombinant DNA reagent | Plasmid pDD268 | (*Dickinson et al., 2015*) | N/A |
| Recombinant DNA reagent | Plasmid pJW1219 | (*Ward, 2015*) | Addgene # #61250 |

*Continued on next page*

*Continued*

| Reagent type (species) or resource | Designation | Source or reference | Identifiers |
|---|---|---|---|
| Recombinant DNA reagent | Plasmid pCFJ90 | Dr Mike Boxem Laboratory | Addgene #19328 |
| Recombinant DNA reagent | Plasmid pPD129.36 (L4440) | Dr Andrew Fire Laboratory | Addgene #1654 |
| Recombinant DNA reagent | Plasmid: pET-21b-*ast-1* | This paper | N/A |
| Recombinant DNA reagent | Plasmid: pET-21b-*unc-86* | (*Zhang et al., 2014*) | N/A |
| Recombinant DNA reagent | Plasmid: *HSNearlyprom::ast-1* | This paper | N/A |
| Recombinant DNA reagent | Plasmid: *HSNearlyprom::hlh-3* | This paper | N/A |
| Recombinant DNA reagent | Plasmid: *HSNearlyprom::ast-1*, *HSNearlyprom::hlh-3* | This paper | N/A |
| Recombinant DNA reagent | Plasmid: *bas-1prom*::ast-1 | This paper | N/A |
| Recombinant DNA reagent | Plasmid: *bas-1prom*::Pet1 | This paper | N/A |
| Recombinant DNA reagent | Plasmid: *cat-4prom*::hlh-3 | This paper | N/A |
| Recombinant DNA reagent | Plasmid: *cat-4prom*::Ascl-1 | This paper | N/A |
| Recombinant DNA reagent | Plasmid: *cat-4prom*::egl-46 | This paper | N/A |
| Recombinant DNA reagent | Plasmid: *cat-4prom*::Insm1 | This paper | N/A |
| Recombinant DNA reagent | Plasmid: *cat-4prom*::Gata2 | This paper | N/A |
| Recombinant DNA reagent | Plasmid: *kal-1prom*::Sall2 | This paper | N/A |
| Sequence-based reagent | oligonucleotides | *Supplementary file 4* | |
| Commercial assay or kit | HisTrap HP Column | GE Healthcare Life Sciences (Marlborough, MA) | Cat#17-5248-01 |
| Commercial assay or kit | QuikChange XL Site-Directed Mutagenesis Kit | Agilent (Santa Clara, CA) | Cat# 200516 |
| Chemical compound, drug | EasyTides Adenosine 5'-triphosphate (ATP [γ−32P]) | Perkin Elmer (Waltham, MA) | Cat#NEG502A250UC |
| Chemical compound, drug | Power Broth Medium | Molecular Dimensions (Maumee, OA) | Cat#MD12-106-1 |
| Chemical compound, drug | Lipofectamine 2000 | Invitrogen (Carlsbad, CA) | Cat#11668027 |
| Chemical compound, drug | Isopropyl-β-D-thiogalacto pyranoside (IPTG) | Acros Organics, ThermoFisher Scientific (Waltman, MA) | Cat#BP1755-100 |
| Chemical compound, drug | Collagenase type IV | Sigma Aldrich | C-5138 |
| Chemical compound, drug | FluorSaveReagent | Merck Millipore (Darmstadt, Germany) | 345789–20 ML |
| Software, algorithm | Gorilla | (*Eden et al., 2009*) | http://cbl-gorilla.cs.technion.a |
| Software, algorithm | R | (Team, 2016) | https://www.r-project.org/ |
| Software, algorithm | Bioconductor | (*Huber et al., 2015*) | https://www.bioconductor.org |
| Software, algorithm | pvclust (R package) | (*Suzuki and Shimodaira, 2006*) | www.sigmath.es.osaka-u.ac.jp shimo-lab/prog/pvclust |

### *C. elegans* strains and genetics

*C. elegans* culture and genetics were performed as described (*Brenner, 1974*). Strains used in this study are listed in *Supplementary file 3*.

### Mouse samples

Animals of C57Bl/6JRccHsd genetic background were housed in an animal care facility with a 12 hr dark/light cycle and had free access to food and water. All experiments were performed according to the animal care guidelines of the European Community Council (86 / 609 / EEC) and to Spanish regulations (RD1201 / 2005), following protocols approved by the ethics committees of the Consejo Superior Investigaciones Científicas (CSIC).

### Generation of *C. elegans* transgenic lines

Gene constructs for *cis*-regulatory analyses were generated by cloning into the pPD95.75 vector. For the identification of the putative binding sites the following consensus sequences were used: ETS: CGGAWR (*Wyler et al., 2016*), GATA: GATA (*Merika and Orkin, 1993*); HLH: CAGAA/ACGTG MatInspector Software (*Cartharius et al., 2005*); INSM: KNNWGSGG (*Breslin et al., 2002*); SPALT: TTGTST (Toker AS 2003) and MatInspector Software (*Cartharius et al., 2005*); POU: WTKCAT (*Weirauch et al., 2014*) and (*Sze et al., 2002*). Mutagenesis was performed by Quickchange II XL site-directed mutagenesis kit (Stratagene, Santa Clara, CA). Reporters for HSN regulatory signature analysis were generated by fusion PCR (*Hobert, 2002*). Generated strains and primers are listed in the *Supplementary file 3* and *4*. For *hlh-3* mutant rescue experiments, the entire coding sequence of *hlh-3* was cloned in front of the heat shock inducible promoter (*hsp16-2*). The transgenic DNA mix was composed by *hlh-3* cDNA (50 ng/µl), together with the co-injection markers *rol-6(su1006)* (50 ng/µl) and *ttx-3::mCherry* (50 ng/µl). For HSN precocious maturation experiments, cDNAs of *ast-1* and *hlh-3* were amplified by PCR and cloned in front of an HSN-specific promoter that drives early expression in the HSN (see promoter sequence below) and the transgenic DNA mix concentrations were the same as above. When both cDNAs were co-injected, we used 25 ng/µl for co-injection markers. For rescue experiments, cDNAs corresponding to the entire coding sequence of *ast-1*, *hlh-3*, *egl-46*, Pet1, Ascl1, Insm1, Sall2 and Gata2 were amplified by PCR and cloned in front of cell-specific promoters: *bas-1prom*, *cat-4prom* and *kal-1prom* (primers in *Supplementary file 4*). The transgenic DNA mix was composed by the DNA of interest [*ast-1* (50 ng/µl in HSN early maturation experiments and 5 ng/µl in HSN rescue experiments), *hlh-3* (50 ng/µl), *egl-46* (50 ng/µl), Pet1 (10 ng/µl), Ascl1 (50 ng/µl), Insm1 (50 ng/µl), Gata2(50 ng/ul) and Sall2 (20 ng/µl)], the co-injection markers *rol-6(su1006)* (50 ng/µl) and *ttx-3::mCherry* (50 ng/µl) and, when necessary, pBlueScript as carrier DNA. DNA was injected into N2 animals and then crossed with their respective mutant strains. *ast-1* and *hlh-3* reporter strains were generated using CRISPR/Cas9-mediated fluorescent protein knock-in, as described in (*Dickinson et al., 2013*; *Dickinson et al., 2015*). For homology arm recombination, we used plasmids containing a self-excising selection cassette: the GFP-containing pJJR82 plasmid (Addgene, Cambridge, MA) in the case of *ast-1*, and the mNeonGreen-containing pDD268 plasmid (*Dickinson et al., 2015*) in the case of *hlh-3*. To target Cas9 to the specific genomic locus, we used the single guide RNA sequence GGGGTGACTATCGATAAAGA for *ast-1*, and GCTATGA TGATCACCAGAAG for *hlh-3*, cloned in the pDD162 (Addgene) and the pJW1219 (Addgene) plasmids respectively. Injection mixes consisted on the Cas9–sgRNA plasmid (50 ng/µl for *ast-1* and 100 ng/µl for *hlh-3*), the repair template (10 ng/µl for *ast-1* and 20 ng/µl for *hlh-3*), and a pharyngeal co-injection marker [2.5 ng/µl pCFJ90 (*Pmyo-2::mCherry*); Addgene].

### Scoring

Scoring and images were performed using 60X objective in a Zeiss Axioplan2 microscope. Lack of GFP signaling was considered OFF phenotype. As we observed no appreciable bias in reporter expression between left and right HSN neurons, percentages were calculated regardless of side. *cis*-regulatory reporter and mutant scoring was performed using young adult worms maintained at 25°C, unless indicated. For *cis*-regulatory analysis a minimum of 30 animals (60 HSN cells) per line were scored. For mutant analysis at least 100 HSN cells, roughly corresponding to 50 animals, were scored for each genotype. For double mutant analysis, we scored young adult worms and we included an extra phenotype category termed 'dim' whenever fluorescence was obviously reduced

but still detectable. For HSN regulatory signature analysis, the three lines showing strongest GFP under the dissecting scope were selected for scoring under the microscope. To prepare figures for publication, images were cropped and rotated, brightness and contrast were adjusted, and maximum intensity projections (where applicable) were performed using FIJI. No other image manipulations were performed.

For wild type TF expression analysis at HSN birth, an *unc-86* fosmid reporter was crossed with the desired TF reporter in order to construct double reporter strains, when possible. UNC-86 is expressed in the HSN after cell cycle exit, approximately 400 min after fertilization and coinciding with embryonic comma stage, which was chosen as analytical time point (*Desai et al., 1988*; *Finney and Ruvkun, 1990*). In the particular case of *hlh-3*, 1 to 2 cell-stage embryos with the endogenous gene tagged were selected and mounted [0 hr post-fertilization (hpf) to 0.8 hpf, respectively], incubated at 25°C and analyzed at different time points. We determined that HLH-3 is initially expressed in the HSN/PHB precursor cell (approximately five hpf) and maintained in the postmitotic HSN. HSN cells were identified relative to nearby landmark cell deaths (*Sulston et al., 1983*). The rest of developmental stages of the worm were identified by standard anatomical features. For *hlh-3* mutant time-specific rescue experiments using the *hsp16-2* promoter, synchronized worms were grown until early L4 larva stage, when they received three heat shock pulses (30 min at 37°C) with 2 hr resting intervals. Animals were analyzed the next morning at young adult stage.

## Statistical analysis for HSN scorings

Data was categorically classified as 'on' or 'off' and the significance of the association was examined using the two tailed Fisher's exact test. For double mutant analysis, 'phenotype' vs. 'no phenotype' was compared and thus, 'dim' and 'off' were considered under the category 'phenotype'. The null hypothesis was that the level of expression in the double mutant would be equal to the product of the levels of expression in single mutants (*Mani et al., 2008*). Whatever statistically deviated from the expected, was considered genetic interaction; Pearson's chi-squared test was used.

## Immunohistochemistry

*C. elegans* serotonin antibody staining was performed using the tube fixation protocol (*McIntire et al., 1992*). Briefly, synchronized young adult hermaphrodites were fixed in 4% paraformaldehyde (PFA) for 18 hr, with β-mercapto-ethanol for another 18 hr, with 1 mg/ml collagenase (Sigma Aldrich, Merk, Darmstadt, Germany) for 90 min and incubated for 24 hr with rabbit anti-5HT antibody (1:5000; Sigma Aldrich). Alexa 555-conjugated donkey anti-rabbit (1:500; Molecular probes) was used as secondary antibody.

For mouse immunohistochemistry, freshly isolated E11.5 embryos from C57Bl/6JRccHsd were fixed by immersion in 4% PFA. Rabbit anti-Sall2 (1:100; Sigma Aldrich), goat anti-Brn2 (1:100; Santa Cruz Biotechnology, Santa Cruz, CA), rabbit anti-5HT (1:5000; Sigma Aldrich) and goat anti-5HT (1:200; Abcam, Cambridge, UK) antibodies were used. As secondary antibodies Alexa 555-conjugated donkey anti-rabbit and anti-goat, and Alexa 488-conjugated donkey anti-rabbit and anti-goat were used (1:600; Molecular probes, Invitrogen, Eugene, OR). Immunofluorescence samples were analyzed and photographed using a confocal TCS-SP8 Leica microscope.

## Promoter sequences required for the generation of transgenic lines

HSN early promoter was generated from *tph-1prom2* in which we incidentally found a point mutation that caused L1 expression:

```
GTAGTAAGCTCCGATGCGTTCCCGTTCATTATTCTTCTTCAATAAATTCGAA
ATCTGACATCATTCTCATCTTTTCCCATCATCACAAGCCGTGGGCTCATTTA
TTCTCCCACGGAAACCATGACAGCAAAAATAAATAGAGTGGCGCCTTATTC
GACTCATTTCGTTTTTTTTTCTCCGGATATTAGATTGTGTGGCAGGCGGCTC
CATTGTATATTcCGaaCCGAATTtttGAAGCACCACGCCATCGGATATCTAAAA
GAGGAGGTGTCTTTGTTTGCGCATAATAAAACAATCAATCAACACAGCAAA
GACCCCTCTCAACCTCATTTCATGATTTTCTTTGGTTTTTAGGTAGCATTGC
TCTCTTCAATCAT
```

* Mutated nucleotides with respect to *tph-1prom2* are indicated in lowercase letters.

## RNAi experiments

RNAi experiments were performed by the standard feeding protocol (*Kamath et al., 2003*). *rrf-3 (pk1426)* background was used to sensitize worms to the RNAi effects. For maintenance experiments, animals were grown under normal food (OP50) until young adult stage. At this stage we scored *tph-1::yfp* and *cat-1::MDM2::gfp* expression in the HSN to confirm that all animals expressed the fluorescent protein and then we transferred animals to RNAi plates with HT115 bacteria (Novagen) transfected with RNAi clones. Worms were incubated at 15°C for 72 hr and then HSN fluorescent expression was scored. For F1 RNAi scoring, we bleached gravid adults in OP50 plates, eggs were allowed to hatch and worms grew in RNAi treated food. We scored their progeny, which had developed under the embryonic effects of RNAi knock down (F1 scoring). The experiment was performed in two independent replicates with similar results. As a negative control the L4440 empty vector was used (pPD129.36, Addgene).

## Electrophoretic mobility assays

Full-length *unc-86* and *ast-1* cDNA into the pET-21b His tag expression vector (EMD Millipore, Merk) were kindly provided be Oliver Hobert. They were transformed into *E. coli* Rosetta2 (DE3) (Novagen) strain. Overexpression was done by first growing the cells at 37°C in LB and Power Broth medium (Molecular Dimensions) respectively, supplemented with 100 µg/ml ampicillin, 100 µg/ml chloramphenicol to OD600 = 0.5–0.6 and then inducing expression with 0.5 mM iso-propyl-b-D thiogalactopyranoside (IPTG, Acros Organics, Thermo Fisher, Waltham, MA) at 37°C for 3 hr or 20°C for 16 hr, respectively.

UNC-86 protein was obtained as previously explained (*Zhang et al., 2014*) with minor changes. Briefly, cells were collected by centrifugation and resuspended in buffer A (100 mM $NaH_2PO_4$, 10 mM Tris [pH 7.5], 10% glycerol) supplemented with 1 mM phenylmethanesulfonyl fluoride (PMSF). Cells were lysed by sonication. Soluble and insoluble fractions were separated by centrifugation and analyzed by SDS/PAGE. Protein was substracted from insoluble fraction as follow: insoluble fraction was resuspended in solubilization buffer (buffer A supplemented with 8 M urea) and loaded on a pre-equilibrated His Trap HP column (GE Healthcare, Chicago, IL). The resin was washed with solubilization buffer supplemented with 10 mM imidazole, and protein was eluted with the same buffer supplemented with 500 mM imidazole. Elution buffer was exchanged by progressive dialysis to 20 mM HEPES [pH 7.5], 100 mM NaCl 10% glicerol, 2 mM $MgCl_2$, and the protein was concentrated by centrifugation up to 1.3 µg/µl and stored at −80°C.

For AST-1 protein, cells were collected by centrifugation and resuspended in buffer B (200 mM MES [pH 6.0], 500 mM NaCl, 2 mM $MgCl_2$, 10% glycerol) supplemented with 1 mM PMSF. Cells were lysed by sonication and soluble proteins were loaded on a His Trap HP column (GE Healthcare) pre-equilibrated with buffer B. The resin was washed with buffer B supplemented with 10 mM imidazole, and protein was eluted with buffer B supplemented with 300 mM imidazole. Eluted fraction was analyzed by SDS/PAGE. Imidazole was removed and protein concentrated by centrifugation up to 0.3 µg/ul, and stored at −80°C.

*egl-18* cDNA was cloned into pcDNA.3 vector followed by His tag sequence and transfected with Lipofectamine-2000 (Invitrogen) in HEK293T cells. HEK293T cells were grown in DMEM 10% FBS. After 24 hr, cells were lysed with the following buffer: 1 mM EDTA, 0.5% Triton, 20 mM β-glicerolP, 0.2 mM PMSF, 100 µM $Na_3VO_4$ and protease inhibitor.

EMSAs were performed incubating UNC-86 and AST-1 proteins in a buffer containing 10 mM Tris [pH 7.5], 50 mM NaCl, 1 mM $MgCl_2$, 4% glycerol, 0.5 mM DTT, 0.5 mM EDTA, 1 µg of poly(dIdC), 6 µg of bovine serum albumin and labeled probes for 20 min at room temperature. For EGL-18, protein extracts were incubated in 20 mM Hepes, 50 mM NaCl, 5 mM $MgCl_2$, 5% glycerol, 1 mM DTT, 0.1 mM EDTA, 1 µg of poly(dIdC), 6 µg of bovine serum albumin and 1 µg anti-6xhistag antibody (Abcam) at 4°C for 30 min. As negative control, anti-GFP antibody (Roche, Basel, Switzerland) was used. Then, labeled probes were added and incubated for 20 min at room temperature. Finally, samples were loaded onto a 6% (37.5:1 acrylamide: bisacrylamide) gel and run at 150 V for 4 hr. Gels were then dried and visualized using Fujifilm FLA-500. Probe sequences are listed in *Supplementary file 4*. Primers were annealed and end-labeled with ATP [γ−32P] (Perkin Elmer, Waltham, MA) using T4 PNK (Thermo Scientific) according to the manufacturer's specifications.

## Bioinformatics analysis

Unless otherwise indicated, all the analyses were performed using R and Bioconductor (*Huber et al., 2015*).

For *C. elegans* regulatory signature analysis, we built PWMs from the functional motifs found in the 5HT pathway genes CRMs (*Figure 5*). Next, we downloaded upstream and intronic gene regions from WormBase version 262 and classified genes in three groups: genes known to be expressed in HSN, genes expressed in neurons and non-neuronal genes, according to WormBase annotations on gene expression and/or belonging to the published neuronal genome (*Hobert, 2013*). PWMs were aligned to genomic sequences and we retrieved matches with a minimum score of 70%. To increase specificity, we removed all matches that did not bear an exact consensus sequence for the corresponding TF family (ETS: YWTCCG, GATA: DGATAD, HLH: SCAGAA, INMS: CCSCWNNM, SPALT: TTGTST, POU: WTKCAT). Then, we performed a sliding window search to find regions that included at least one match for four or more of the 6 TF types. Windows were separated according to the number of different motifs that they bore (4, 5 or 6), and then overlapping regions were merged. Embryonic stem cell enhancers median size has been reported to be around 800 bp (*Parker et al., 2013*); therefore, the initial search was performed with a maximum length restriction of either 600, 700 or 800 bp. Differences between HSN-expressed genes and other gene groups was greater when the maximum length was set to 700 bp, thus we kept this maximum window length for the rest of the analyses. To assess enrichment in signature in HSN-expressed genes we sampled 10,000 groups of 96 genes that (1) had not previously been reported to be expressed in the HSN, (2) at least one ortholog had been described in other *Caenorhabditis* species (*C. briggsae*, *C. japonica*, *C. remanei* or *C. brenneri*), and (3) such that their upstream and intronic regions were similar in length, on average, to those of the HSN-expressed genes (Mann-Whitney U test, p-value>0.05). We compared the distribution of the proportion of genes with signature (4, 5 or 6 different motifs) in these groups to the HSN-expressed gene group. We consider the enrichment in signature to be significant when the percentile of the HSN-expressed group is above 95. In order to assess signature conservation, we performed a similar motif search using other nematode genomes also available from WormBase (*C. briggsae*, *C. japonica*, *C. remanei*, *C. brenneri*) and we considered the signature to be conserved if HSN regulatory windows were found in all orthologous genes, at least 4, 5 or 6 motifs for 4, 5 and 6-motif *C. elegans* windows.

Gene ontology analysis was performed using GOrilla software, using *C. elegans* coding genome (19.276 genes) as control list (*Eden et al., 2009*).

For hierarchical clustering, we used curated data from WormBase (*Hobert et al., 2016*) to generate a matrix with gene expression profiles for the 118 *C. elegans* hermaphrodite anatomical neuronal classes. Pan-neuronal genes and neurons in which less than 30 genes had been reported to be expressed were excluded. We built a similar matrix with mouse gene expression data from RNA-seq experiments, either from adult raphe nuclei divided into different rhombomeres (R1Dorsal, R1 Medial, R2, R3, R5, R6) (*Guillemot and Hassan, 2017*) or from cortical neurons that served as negative control (*Molyneaux et al., 2015*)). To transform the quantitative RNA-seq data into a presence-absence binary matrix. We considered values above 19 CPM as present and values below that threshold as absent because this cut-off produces a list of approximately 7000 expressed genes in each Raphe sample (roughly a third of the genome that is what is being estimated as expressed in a given cell type). Nevertheless, results were consistent in all conditions when considering cutoffs ranging from 9 to 140 CPM, after which HSN-raphe cluster robustness started to decline (low AU and BP values, not shown).

To assign mouse ortholgs to *C. elegans* genes, we combined orthology relationships between mouse and worm genes annotated in the ENSEMBL database and worm-human orthology relationships reported in *Shaye and Greenwald (2011)*. In the last case, we used ENSEMBL database again to assign mouse orthologs to human genes. In (*Molyneaux et al., 2015*)), ENSEMBL, OrthoMCL, InParanoid and Homologene methods are combined to identify orthologs. Thus, we combined both sources to have a wider coverage of orthology relationships than using ENSEMBL or (*Molyneaux et al., 2015*) data alone. Worm genes without any mouse ortholog and genes that were not expressed in any worm neuron were removed. Whenever a worm gene had more than one mouse ortholog, it was duplicated in the worm data set. For hierarchical clustering, this binary matrix containing mouse and worm expression data was fed to the pvclust function in the pvclust R

package (*Shimodaira, 2002*), which uses a bootstrapping technique to calculate p-values for each cluster, the AU and BP values (*Shimodaira, 2002*). Parameters were set as follows: method.hclust = 'average', method.dist= 'binary', nboot = 10,000, r = seq(0.5, 1.4, by = 0.1). The standard error of the PV and AU values was approximately 0.1% for most clusters, including the HSN-raphe cluster. Also, as a control, 100 random sets of 96 expressed genes (the same number of genes that are expressed in the HSN) were generated from the worm gene pool. Each random set contained the four 5HT pathway genes (*tph-1, cat-1, cat-4* and *bas-1*) plus 92 randomly picked genes from the genes expressed in *C. elegans* neurons. This data set was merged with mouse raphe nuclei expression profile and pvclust was run as before.

## Acknowledgements

We thank CGC (P40 OD010440) for providing strains. Dr Oliver Hobert, as initial experiments were performed in his laboratory and for sharing unpublished data. Dr Oscar Marín and Dr Beatriz Rico labs for scientific discussion. Elia García, Vicent Puig and Benito Alarcón for technical help. Dr Alberto Marina, Rafa Ciges and Dr Marta Casado for help with protein purification and EMSAs, Dr Iñaki Comas for sharing server space, Dr Mario Sendra for statistics advice, Life Science Editors for manuscript editing and Dr Esteban Mazzoni, Dr Luisa Cochella, Dr Inés Carrera, Dr Carlos Estella, Dr José Penadés and Dr Oscar Marín for comments on the manuscript.

## Additional information

### Funding

| Funder | Grant reference number | Author |
| --- | --- | --- |
| Ministerio de Economía y Competitividad | SAF2014-56877-R | Carla Lloret-Fernández<br>Miren Maicas<br>Carlos Mora-Martínez<br>Ángela Jimeno-Martín<br>Laura Chirivella<br>Nuria Flames |
| European Research Council | ERC Stg 2011-281920 | Carla Lloret-Fernández<br>Miren Maicas<br>Carlos Mora-Martínez<br>Ángela Jimeno-Martín<br>Laura Chirivella<br>Nuria Flames |
| Generalitat Valenciana | ACIF/2013/087 | Carla Lloret-Fernández |
| Ministerio de Educación, Cultura y Deporte | FPU14/02651 | Carlos Mora-Martínez |
| Generalitat Valenciana | ACIF/2015/398 | Ángela Jimeno-Martín |

The funders had no role in study design, data collection and interpretation, or the decision to submit the work for publication.

### Author contributions

Carla Lloret-Fernández, Miren Maicas, Formal analysis, Investigation, Visualization, Writing—original draft, Writing—review and editing; Carlos Mora-Martínez, Software, Formal analysis, Investigation, Visualization, Methodology, Writing—original draft, Writing—review and editing; Alejandro Artacho, Software, Methodology; Ángela Jimeno-Martín, Formal analysis, Investigation, Writing—original draft, Writing—review and editing; Laura Chirivella, Investigation, Visualization; Peter Weinberg, Resources; Nuria Flames, Conceptualization, Supervision, Funding acquisition, Visualization, Methodology, Writing—original draft, Project administration, Writing—review and editing

### Author ORCIDs

Nuria Flames (iD) http://orcid.org/0000-0003-0961-0609

### Ethics

Animal experimentation: All experiments were performed according to the animal care guidelines of the European Community Council (86 / 609 / EEC) and to Spanish regulations (RD1201 / 2005), following protocols approved by the ethics committees of the Consejo Superior Investigaciones Científicas (CSIC).

### Decision letter and Author response

Decision letter https://doi.org/10.7554/eLife.32785.035
Author response https://doi.org/10.7554/eLife.32785.036

---

## Additional files

### Supplementary files

• Source data 1. Raw scoring data and statistical analysis. Related to *Figure 1*, *Figure 2*, *Figure 3*, *Figure 8*.
DOI: https://doi.org/10.7554/eLife.32785.028

• Supplementary file 1. Phenotypic characterization of additional alleles of the HSN TF combination. Related to *Figure 1*.
DOI: https://doi.org/10.7554/eLife.32785.029

• Supplementary file 2. HSN expressed, neuronal genome and non-neuronal genome gene lists. Related to *Figure 7*.
DOI: https://doi.org/10.7554/eLife.32785.030

• Supplementary file 3. Strains. Related to Materials and methods.
DOI: https://doi.org/10.7554/eLife.32785.031

• Supplementary file 4. Primers. Related to Materials and methods.
DOI: https://doi.org/10.7554/eLife.32785.032

• Transparent reporting form
DOI: https://doi.org/10.7554/eLife.32785.033

---

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
