## [Decision Letter]

[Editors’ note: this article was originally rejected after discussions between the reviewers, but the authors were invited to resubmit after an appeal against the decision.]

Thank you for submitting your work entitled "A combinatorial transcription factor signature defines the serotonergic neuron regulatory landscape" for consideration by *eLife*. Your article has been favorably evaluated by a Senior Editor and three reviewers, one of whom is a member of our Board of Reviewing Editors. The reviewers have opted to remain anonymous.

Our decision has been reached after consultation between the reviewers. Based on these discussions and the individual reviews below, we regret to inform you that your work will not be considered further for publication in *eLife* in its current format.

We have opted to include the full reviews below. As you will see, all reviewers are potentially interested in the work, but all ask for extensive revisions and additional experiments to support the conclusions. Although we are unable to move forward with the current version of the manuscript given the extent of the requested revisions, if you agree with the reviewers and are willing and able to do the extensive work outlines below we do encourage you to consider *eLife* again for submission of a revised manuscript once the issues raised below have been addressed. The manuscript will be considered a new submission. If, on the other hand, you do not agree with the reviewers, please do not do experiments you don't consider worthwhile or merited, but instead submit your manuscript elsewhere.

*Reviewer #1:*

Positives:

- The fact that the authors characterize a different TF regulatory mechanism for HSN than what has been described by the Hobert lab for the majority of other neurons

- The work is extensive and detailed (but see below) and the conservation with mouse serotonergic neuron regulatory mechanisms is encouraging.

Negatives:

- I am bothered by the strong conclusions the authors draw from looking at single alleles of genes. Given the variable penetrance of the observed expression phenotypes, some of this could easily arise from background effects.

- It wasn't clear to me that they had carefully looked at developmental time courses for expression in the various single and double mutant backgrounds. It looked like only adults were scored. In other words, it is possible that some of these regulators may play a role in defining the temporal patterns of expression? Ectopic temporal expression could also disrupt development.

- I didn't understand how mutating *hlh* binding sites leads to an effect if HLH-3 is not expressed in the adult.

- It seems important to look at expression upon mutating multiple TF binding sites and not just at double mutants for the TFs.

- Finally, I am just not sure how much we've learned from looking at this single neuron. Perhaps this is not a fair criticism but given the data, it's not even clear to me that the authors can make very strong statements about what they've learned.

*Reviewer #2:*

Overall this is a nice body of work. The two most interesting points are relevant to:

1) Transcriptional regulation, enhancer function: The identification of another good example of a TF collection mode of regulation; that is a group of TFs essential for the regulation of a shared set of enhancers and genes, whereby the input of each TF on different enhancers within the set can vary from direct to indirect recruitment- such that the binding sites of all TFs don't have to be present on all enhancers in the set;

2) Developmental and evolutionary standpoint: This study identifies the major TFs essential for serotonergic neurons in *C. elegans*, a set of 6 factors with very different DNA binding domains, which have a deeply conserved role in serotonergic neuron development.

In reading the paper, I found it very strange to talk about regulatory codes (subsection “Transcription factors from six different families are required for HSN 3 neuron terminal differentiation”) when describing Figure 1 and 2. At least to me, codes refer to something in the DNA sequence, while the first two figures describes the proteins involved, their loss-of-function phenotypes, and their expression. For example, – "…this set of six TFs that we refer as the HSN TF code". As you have already referred to the TF collective in the Introduction, why not already call it a TF collective here rather than code or simply 'the HSN TFs' and then define it as a HSN TF collective later as you currently do.

The analysis of enhancer activity in the double mutants showing synergy etc. is very nice – but some of these embryos are quite messed up. To avoid pleiotropic functions that could lead to secondary effects have you tried making double mutants in the TFBSs of the analyzed enhancers? This would be a very nice complement to Figure 5.

The motif prediction section is very interesting. Given your genetic dissection showing that for a given TF, its TFBS are essential for some enhancers and indispensable for others in keeping with the TF collective mode of regulation, I was surprised that the authors required the presence of all 6 TFBSs. I realize that you wanted to make the screen as stringent as possible – and even with all 6 conserved only ~35% of the predictions are expressed in the correction tissue (as you discuss very openly). Given the apparent plasticity in the system, it would be very interesting to do the same computational analysis searching for elements that contain any 5 and any 4 of the 6 TFBS. Does this decrease your enrichments or not make any difference (as the collective model would propose)?

- "Of note, 5HT pathway gene CRMs are not completely independent as in some cases they partially overlap (Figure 3)." Just because the elements have overlapping expression doesn't mean that they are not acting independently. By independent do you mean each enhancer functions as an independent autonomous unit? Or that they are regulated by independent (unique) sets of TFs?

The deep homology of the system is really interesting, and the rescue of the worm mutants with the human factors is very impressive. Actually, I suggest to consider moving this down (Figure 7D) to the end of the figure. Start with the expression of the mouse genes, then the expression comparison to the mouse 5HT raphe suggesting deep homology specifically to the HSN neurons. To test this deep homology you then determined if the mouse TFs could rescue the *C. elegans* mutants, suggesting that the underlying regulatory network is also conserved.

*Reviewer #3:*

In this manuscript, the authors use *C. elegans* to study transcription factor combinations that are responsible for the differentiation of HSN, a specific class of serotonergic neurons. They identify six transcription factors based on their egg laying phenotype, as well as the absence of serotonin production in mutant cells. They argue that these transcription factors act in parallel and synergistically to regulate the serotonergic regulatory landscape. They study the enhancers of genes involved in the serotonin biosynthetic pathway where they identify the binding sites for these six transcription factors. They use this information to try to find more genes that are regulated by these transcription factors. Although they manage to do so, they have a very high number of false positives. Then, they use the mouse orthologs of these transcription factors to see if they can drive HSN differentiation in worms. Finally, using transcriptomic data they conclude that the HSN neurons are homologous to the mouse raphe serotonergic neurons.

The manuscript has a number of interesting results, but there is a lot of missing information. The interpretation of the results is often exaggerated and many conclusions are extrapolated from patchy data.

- The title is misleading. The authors advertise the TF combination as regulating serotonergic neurons, while they are only studying one serotonergic type. In fact, the other serotonergic neurons are minimally affected in these TF mutants.

- The authors are obviously missing a number of important transcription factors that participate to the differentiation of these neurons. Therefore, this makes most of their downstream results (that is synergy and epistasis) hard to interpret. The way the authors came up with their candidates is very biased towards genes that do not have embryonic lethal phenotypes; in fact, most transcription factors are re-used in different contexts, including embryogenesis.

- Very importantly, the authors observe many non serotonin-related genes being affected in the TF mutants. This means that they are obviously also interfering with the differentiation of the HSNs! Some genes like *kcc-2* and *lgc-55* are affected equally to the serotonergic pathway genes, while serotonergic pathway genes, such as *cat-4*, are affected less than some terminal features. However, the authors conclude: "This indicates that HSN partially differentiates, but fails to activate expression of the complete HSN transcriptome". An equally possible conclusion is that they are affecting the differentiation of HSNs, but since they are missing many TFs from their collection, they only affect it partially.

- How is HLH3 important for the serotonergic identity when it is not expressed in the adults? The neurons do continue to synthesize serotonin in adult worms. It could be acting as a pioneer factor, or it could be relaying its activity to a downstream transcription factor, but the authors never address this.

- In Figure 2E, can the authors really decide on the fate maintenance based on one marker? Moreover, most of the TFs have a very minimally (although significantly) penetrant effect. Do they believe that the egg-laying defect is because of this effect? Could it be that this mutant is affecting other cell types that are causing the egg-laying defects?

- As stated earlier, Figure 5 is quite confusing. It shows complex regulatory interactions that cannot be put in a model, as the data are incomplete. Moreover, the authors are testing only TFs with low penetrance and are extrapolating for the whole series.

[Editors’ note: what now follows is the decision letter after the authors submitted for further consideration.]

Congratulations, we are pleased to inform you that your article, "A combinatorial transcription factor signature defines the HSN serotonergic neuron regulatory landscape", has been accepted for publication in *eLife*.

This is a detailed analysis of the TF network that specifies the differentiated characteristics of a single serotonergic neuron type in *C. elegans*. The authors show that a "TF collective" comprised of 6 TFs specify this neuron's fate in contrast to the master selector model. Mammalian orthologs are able to substitute for the worm TFs and are expressed in 5-HT neurons, suggesting a deep conservation.

Although the first version of this manuscript was rejected, the authors performed a large number of experiments and both reviewers have concurred that they have addressed all the previous comments very well.

A comment from reviewer 2 that you may want to take into consideration in your final manuscript: Given the now very strong focus and evidence for the TF collective model in this system, why not go the whole hog, and state this in the title: A transcription factor collective defines the HSN serotonergic neuron regulatory landscape

---

## [Author Response]

Reviewer #1:[…] Negatives:- I am bothered by the strong conclusions the authors draw from looking at single alleles of genes. Given the variable penetrance of the observed expression phenotypes, some of this could easily arise from background effects.

We thank the reviewer for this important remark. We have now included the quantification of at least two alleles for each member of the HSN TF collective. For each allele the expression of at least two 5HT pathway gene reporters and 5HT staining defects are analyzed. We also include RNAi phenotypes for *tph-1* and *cat-1* expression for each HSN TF collective member. These results, described now in the text (subsection “Transcription factors from six different families are required for HSN neuron terminal differentiation”, third paragraph) and in Figure 1—figure supplement 1 and Supplementary file 1, show that HSN phenotypes are due to mutations in the TFs and not to background effects.

- It wasn't clear to me that they had carefully looked at developmental time courses for expression in the various single and double mutant backgrounds. It looked like only adults were scored. In other words, it is possible that some of these regulators may play a role in defining the temporal patterns of expression? Ectopic temporal expression could also disrupt development.

We did not include this data in the original submission for simplicity and we realized now that it is an important piece of information. We thank very much the reviewer for pointing this out. As stated by the reviewer, dysregulation of temporal expression is also a phenotype important to consider.

The onset of expression of all analyzed reporter strains is around L4 larval stage, when HSN neuron matures (See Figure 1: *tph-1, bas-1, cat-1, cat-4, kcc-2, lgc-55, ida-1*, etc.). The only exception being *kal-1*, which is expressed already at larval stage 1 (L1). None of the single mutants show precocious expression of any of the terminal features when we analyzed expression at first larval stage. This observation suggests that the members of the HSN TF collective act mainly as activators of expression in the onset of differentiation but do not act as repressors of ectopic expression before HSN maturation. We have now included this observation as a comment in the revised version of the manuscript:

'HSN neuron is born embryonically and remains in a quiescent undifferentiated state until late fourth larval stage (L4), when it activates the expression of most effector genes, including the 5HT pathway genes. We did not observe precocious expression of any of the analyzed terminal features in any of the single mutant backgrounds suggesting that the HSN TF combination acts mainly as activator of transcription'

- I didn't understand how mutating hlh binding sites leads to an effect if HLH-3 is not expressed in the adult.

We apologize for not being clear enough on the role of HLH-3, as also pointed by reviewer #3. In the revised version of the manuscript we have tried to clarify this issue with additional experiments and re-writing of the section: detailed analysis of *hlh-3* expression by GFP fusion to HLH-3 in the endogenous locus (Figure 2—figure supplement 1A) shows that, post-embryonically, HLH-3 starts being expressed in the HSN at third larval stage and its expression is highest at late third larval stage and early forth larval stage, this expression shortly precedes and overlaps onset of expression of most HSN effector genes (Figure 2—figure supplement 1C). We also demonstrate that *hlh-3* postembryonic expression in *hlh-3* mutants is sufficient to rescue *tph-1* expression defects (Figure 2C). Taking into account that we found functional bHLH binding sites in *tph-1* and *cat-1* CRMs (Figure 5), altogether our data supports the hypothesis that *hlh-3* function is directly required to initiate expression of some HSN effector genes. A role for TFs exclusively in initiation but not maintenance has been previously reported. For example, embryonic *ttx-3* expression in AIY neuron is dependent on the TF REF-2 but its maintenance of expression does not require REF-2 but TTX-3/CEH-10 activity (Bertrand and Hobert, 2009). There are also examples in mouse of TFs required for establishment but not maintenance of cell fate (Bettess et al., 2005, Hoogenkamp et al., 2009, Martins et al., 2005) or required for maintenance but not initiation of expression (Galceran et al., 2001, Harms et al., 2014).

HLH-3 belongs to the Asc proneural factor family and is ortholog to mouse Ascl1 (Figure 2—figure supplement 1D). Ascl1 is expressed in serotonergic progenitors and early postmitotic serotonergic neuroblasts. Besides its role in neuronal progenitors specification it has well-demonstrated roles inducing neuronal differentiation in serotonergic and other neuronal types (Pattyn et al., 2004, Sommer et al., 1995). Ascl1 is also part of the TF combinations used to induce serotonergic fate from human fibroblasts (Vadodaria et al., 2015, Xu et al., 2015). Similar to *hlh-3*, Ascl1 expression is downregulated from mature neurons. Temporal dysregulation of Ascl1 activity by sustained expression or inhibition of degradation leads to neuronal differentiation defects (Imayoshi et al., 2013, Urbán et al., 2016). Similarly, we observed that precocious and sustained expression of *hlh-3* leads to HSN differentiation defects (Figure 2D). *hlh-3* function shows additional shared features to Ascl1 like regulation of expression of panneuronal features (see *rab-3* expression defects Figure 1E) or being an upstream regulator of *egl-46* (Figure 3A) (Ascl1 directly regulates Insm1, ortholog of *egl-46*) (Castro et al., 2011). Finally, Ascl1 can substitute *hlh-3* function in HSN differentiation (Figure 8F). Altogether our new data suggest that in the HSN TF collective *hlh-3* acts as a proneural gene in the induction of HSN terminal fate. Ascl1 has been shown to act as a pioneer factor to facilitate binding of additional TFs (Wapinski et al., 2013). It would be very interesting to address if HLH-3 can also bind closed chromatin. We have not addressed this issue in the manuscript because we think it is out of the scope of the paper, the main message of the paper is the identification and characterization of HSN TF collective as a whole, our ability to predict HSN enhancers and the phylogenetic conservation. We believe that a more detailed analysis on HLH-3 function as proneural factor is a project on its own for future manuscripts.

This data is now explained in the subsection “AST-1 acts as temporal switch for HSN maturation” and Figure 2 and (Figure 2—figure supplement 1).

- It seems important to look at expression upon mutating multiple TF binding sites and not just at double mutants for the TFs.

We thank reviewer #1 for this suggestion, also proposed by reviewer #2. We have now performed these experiments that are summarized in Figure 6—figure supplement 1. Double *cis* mutant analysis was limited to the few TFBS that produce partial loss of expression upon mutation. This reduced the number of possible constructs to four. We analyzed three of these four combinations and found that they produce epistatic effects but not synergistic effects. We still don´t know why we did not find synergy (even in genes for which corresponding double trans mutants did show synergy), one possible explanation is that, considering the function of HSN TF collective as a whole, TFBS mutations could be more easily compensated than mutations in the corresponding trans-activating factors.

To widen our analysis for genetic interactions in the HSN TF collective and trying to minimize TF pleiotropic effects we also performed several cis-trans double combinations (Figure 6I). In this context we found synergy as well as additivity and suppression effects.

Altogether our double mutant analysis (both cis and trans) is in agreement with the HSN TF collective model that predicts extensive synergy among factors and supports the idea of physical interactions or facilitation of DNA accessibility among TFs. It also shows the great flexibility of the system as specific CRM structure and context determines double TF effects in the regulation of each particular gene.

This information is now in the subsection “HSN TF collective shows enhancer-context dependent synergistic relationships” and in Figure 6 and Figure 6—figure supplement 1.

- Finally, I am just not sure how much we've learned from looking at this single neuron. Perhaps this is not a fair criticism but given the data, it's not even clear to me that the authors can make very strong statements about what they've learned.

We apologize for not being clear enough about the advances provided by our work, we have tried to clarify them in the new Discussion, main important advances provided by our work:

1) As pointed earlier by reviewer #1, our work on the HSN logic shows a different picture, much more complex than previously presented for other *C. elegans* neurons. We think this is not due to an intrinsic difference in the HSN, but to the exhaustive analysis we have performed. Describing this complexity is important because the data available so far created the false impression that *C. elegans* cell fate regulation was different and much simpler than in other organisms and thus could follow different rules see for example review (Holmberg and Perlmann, 2012). (This concept is explained in the first section of the Discussion).

2) Three models have been proposed to explain enhancer function: the enhanceosome model, billboard model and TF collective model. It is still a matter of debate which model better represents TF way of action. There are several beautiful examples that fit the TF collective model mainly, but not only, in *Drosophila* embryo. To our knowledge our work is the first characterization in great detail of the action of a TF collective model in the neuronal subtype terminal differentiation. Moreover our detailed examination on the HSN TF collective function has allowed us to describe a high degree of flexibility as well as extensive synergistic relationships further supporting the TF collective model. (This concept is also explained in the first section of the Discussion).

3) Despite great efforts, we still do not understand why specific regions in the DNA constitute functional enhancers. This lack of understanding precludes us from identifying *cis*-regulatory modules only based on DNA sequence. Although we are still far from reaching this goal, our work shows that the complexity of the HSN TF collective, composed by at least six members (and most likely by additional TFs) impose a regulatory signature that is specific enough to identify enhancers only from DNA sequence. Our understanding is still incomplete, what produces a high number of false positives in our search, nevertheless, we think it provides an important proof of concept that could help in the understanding of enhancer selection. (Explained in the second section of the Discussion).

4) Finally, our work shows an unexpected degree of homology between the HSN and the mouse raphe. Both terminal features as well as TF regulatory network seem shared between these two populations. From an evolutionary point of view we think this finding is very relevant and suggests that eumetazoans might share deep homology in additional neuron types. (Explained in the third section of the Discussion).

In addition to the response to specific reviewer #1 comments, we have also included in the revised manuscript:

1) additional support for maintenance role of UNC-86, AST-1, SEM-4, EGL-46 and EGL-18 by RNAi and temperature sensitive alleles (Figure 3C and Figure 3—figure supplement 1);

2) Role of AST-1 as a temporal switch to induce HSN differentiation (Figure 2);

3) Additional bioinformatics analysis for HSN regulatory windows with only five or only four motifs (Figure 7—figure supplement 2) and

4) Regulation of expression of identified HSN active enhancers by *unc-86* (motifs (Figure 7—figure supplement 3).

Reviewer #2:[…] In reading the paper, I found it very strange to talk about regulatory codes (subsection “Transcription factors from six different families are required for HSN 3 neuron terminal differentiation”) when describing Figure 1 and 2. At least to me, codes refer to something in the DNA sequence, while the first two figures describes the proteins involved, their loss-of-function phenotypes, and their expression. For example, "…this set of six TFs that we refer as the HSN TF code". As you have already referred to the TF collective in the Introduction, why not already call it a TF collective here rather than code or simply 'the HSN TFs' and then define it as a HSN TF collective later as you currently do.

We thank reviewer #2 for this very constructive remark. We now realize the term 'code' is confusing and not properly used in the text. As the term 'TF collective' entails the direct action of TFs upon the co-regulated genes we decided to only use HSN TF collective after determining functional binding sites in the HSN CRMs. We have now eliminated the expression 'HSN TF code' from the text and replace it by the 'HSN TF combination' that we think is more appropriate.

The analysis of enhancer activity in the double mutants showing synergy etc. is very nice – but some of these embryos are quite messed up. To avoid pleiotropic functions that could lead to secondary effects have you tried making double mutants in the TFBSs of the analyzed enhancers? This would be a very nice complement to Figure 5.

We thank reviewer #2 for this suggestion, which is also proposed by reviewer #1. As stated by reviewer #2 double mutants can show enhanced secondary effects, indeed a clear example is that although both *egl-18(ok290)* and *ast-1(ot417)* are viable double mutants are embryonic lethal. Following reviewers suggestions we have now performed these experiments that are summarized in Figure 6—figure supplement 1. Double cis mutant analysis was limited to the few TFBS that produce partial loss of expression upon mutation. This reduced the number of possible constructs to four. We analyzed three of these four combinations and found that they produce epistatic effects but not synergistic effects. We still don´t know why we did not find synergy (even in genes for which corresponding double trans mutants did show synergy), one possible explanation is that, considering the function of HSN TF collective as a whole, TFBS mutations could be more easily compensated than mutations in the corresponding trans-activating factors.

To widen our analysis for genetic interactions in the HSN TF collective and trying to minimize TF pleiotropic effects we also performed several cis-trans double combinations (Figure 6I). In this context we found synergy as well as additivity and suppression effects.

Altogether our double mutant analysis (both cis and trans) is in agreement with the HSN TF collective model that predicts extensive synergy among factors and supports the idea of physical interactions or facilitation of DNA accessibility among TFs. It also shows the great flexibility of the system as specific CRM structure and context determines double TF effects in the regulation of each particular gene.

This information is now in the subsection “HSN TF collective shows enhancer-context dependent synergistic relationships” and in Figure 6 and Figure 6—figure supplement 1.

The motif prediction section is very interesting. Given your genetic dissection showing that for a given TF, its TFBS are essential for some enhancers and indispensable for others in keeping with the TF collective mode of regulation, I was surprised that the authors required the presence of all 6 TFBSs. I realize that you wanted to make the screen as stringent as possible – and even with all 6 conserved only ~35% of the predictions are expressed in the correction tissue (as you discuss very openly). Given the apparent plasticity in the system, it would be very interesting to do the same computational analysis searching for elements that contain any 5 and any 4 of the 6 TFBS. Does this decrease your enrichments or not make any difference (as the collective model would propose)?

We thank reviewer #2 for this suggestion. As pointed by the reviewer, we performed our initial analysis as stringent as possible to try to get the 'best possible prediction'. However, also as stated by reviewer, TF collective model predicts that in some contexts, specific TFBS are dispensable. Our lack of understanding of the details in the organization and disposition of the specific TFBS makes yet impossible to predict in which specific contexts one TFBS is absolutely required or just dispensable.

We have performed similar prediction analysis considering regulatory windows with any five or any four motifs and compared them to the original six -motif analysis. Our results are summarized in the subsection “The HSN regulatory signature allows de novo identification of HSN expressed genes” and in Figure 7—figure supplement 2. In summary, we find that in contrast to our analysis with 6 motifs, only five or only four motif windows are not enriched in genes known to be expressed in HSN compared to random (Figure 7—figure supplement 2A, B). Additionally, inclusion of only five motif regulatory windows (≥5 motif) or both only five and only four motif regulatory windows (≥4 motif) decreases the enrichment in the neuronal genome compared to non neuronal genome and the GO term enrichment on HSN related terms (Figure 7—figure supplement 2C-E). Altogether, our data shows that the most prevalent mark of HSN expressed genes is the regulatory signature with all six TFBS. Even if some HSN active enhancers lack predicted binding sites for any of the members of the HSN TF collective, at the genomic level, including enhancers with missing TFBS abolishes cell type specificity.

- "Of note, 5HT pathway gene CRMs are not completely independent as in some cases they partially overlap (Figure 3)." Just because the elements have overlapping expression doesn't mean that they are not acting independently. By independent do you mean each enhancer functions as an independent autonomous unit? Or that they are regulated by independent (unique) sets of TFs?

We thank reviewer #2 for this observation. As very well pointed, the fact that enhancers partially overlap does not necessary imply they share some kind of regulation. We now realized this term sentence might be confusing, we have modified the text and omitted the term independent:

'Of note, 5HT pathway gene CRMs in some cases partially overlap (Figure 4)'.

The deep homology of the system is really interesting, and the rescue of the worm mutants with the human factors is very impressive. Actually, I suggest to consider moving this down (Figure 7D) to the end of the figure. Start with the expression of the mouse genes, then the expression comparison to the mouse 5HT raphe suggesting deep homology specifically to the HSN neurons. To test this deep homology you then determined if the mouse TFs could rescue the C. elegans mutants, suggesting that the underlying regulatory network is also conserved.

We thank the reviewer for this excellent suggestion that improves the flow of the manuscript, we have re-arranged the data from the text and figure accordingly (New Figure 8).

Reviewer #3:[…] The manuscript has a number of interesting results, but there is a lot of missing information. The interpretation of the results is often exaggerated and many conclusions are extrapolated from patchy data.- The title is misleading. The authors advertise the TF combination as regulating serotonergic neurons, while they are only studying one serotonergic type. In fact, the other serotonergic neurons are minimally affected in these TF mutants.

We apologize if the title seemed misleading for the reviewer. As pointed, most of our work is specifically on *C. elegans* HSN serotonergic neuron and not on all *C. elegans* serotonergic subpopulations. The reason why we omitted HSN from the title is that we provide evidence for deep homology between *C. elegans* HSN and mouse raphe serotonergic neurons. We have now modified the title to include the term HSN:

'A combinatorial transcription factor signature defines the HSN serotonergic neuron regulatory landscape'

- The authors are obviously missing a number of important transcription factors that participate to the differentiation of these neurons. Therefore, this makes most of their downstream results (that is synergy and epistasis) hard to interpret.

We completely agree with reviewer #3 that additional unidentified TFs are part of the HSN TF collective, moreover, other TFs are known to be required for earlier events of HSN development such us lineage specification, we have not considered these TFs in our analysis as we are focused specifically in the last steps of HSN maturation. However, we respectfully disagree with the reviewer's claim that, because not all components of the HSN TF collective have been identified, the conclusions from our data are hard to interpret. Biological research is often based on the dissection and analysis of specific parts of complex systems to understand their function. This strategy has been historically very successful to advance our understanding of complex processes. In this context, synergy refers to the relationship between two factors, independently of how many additional factors are involved. Many examples of synergistic relationships have been reported in different organisms in which only a few components of the whole network were considered (see as examples for *C. elegans* (Chen et al., 2013, Doitsidou et al., 2013, Hsin and Kenyon 1999, Kratsios et al., 2017, Ogg et al., 1997, S Lichtsteiner 1995, Sawa et al., 2000, Sommermann et al., 2010, Wang et al., 2014, Zhang et al., 2013) and for other systems: (Costa et al., 2013, Goldstein et al., 2017, Kottakis et al., 2016, Mazzoni et al., 2013, Rubtsova et al., 2013, Stuart et al., 2014)). All these publications provided useful insights in biology.

More specifically, there are other examples of identified TF collective functions in which also additional TFs are missing and nevertheless that provide important insights into the mechanisms of regulation of gene expression (Junion et al., 2012, Uhl et al., 2016).

In summary, we believe that our data, both *cis*-regulatory analysis and double mutant analysis, provides strong evidence that the HSN terminal differentiation fits best the TF collective model of transcriptional regulation in which TFs act together to broadly regulate terminal fate (reviewed in Spitz and Furlong 2012): they interact in some contexts, what can lead to TF recruitment even in the lack of a functional TFBS and accordingly they can show synergistic relationships.

The way the authors came up with their candidates is very biased towards genes that do not have embryonic lethal phenotypes; in fact, most transcription factors are re-used in different contexts, including embryogenesis.

We completely agree with reviewer #3 in that most TFs, including the HSN TF collective, are pleiotropic. All members of the HSN TF collective are broadly expressed during embryogenesis and have other reported functions either in specification of other neuronal types or even in non-neuronal tissues. However, we would like to clarify that we have not limited our analysis to TFs that are not required for early steps in development. This is precisely the case of *ast-1*: null *ast-1* alleles are L1 larval lethal. However, forward genetic screens and the isolation of hypomorphic mutations allows for the identification of viable alleles in genes that code for TFs that are required for viability, and *ast-1* is one of the many examples. Just to mention another case, in the past we isolated a *ceh-43* mutant allele that contains a regulatory mutation that specifically affects *ceh-43* expression in CEPs neurons, this mutant allele is viable while null alleles of *ceh-43* are embryonic lethal. The isolation of this mutation was key in identifying a specific role of *ceh-43* in dopaminergic specification (Doitsidou et al., 2013).

In summary, 1) we are aware, as we state in the manuscript several times, that other components of the HSN TF collective are yet to be identified. These additional TFs may or may not be required for embryonic development. Nevertheless, this fact does not affect the conclusions driven from our data, 2) a member of the HSN TF collective is required for worm viability (*ast-1*), demonstrating that we do not exclude from our analysis genes required for embryonic development.

We have now included a sentence in the Discussion section acknowledging the possibility that the additional components of the HSN TF collective could have a role in embryogenesis (subsection “Regulation of C. elegans neuron specification is unexpectedly complex”, last paragraph).

- Very importantly, the authors observe many non serotonin-related genes being affected in the TF mutants. This means that they are obviously also interfering with the differentiation of the HSNs! Some genes like kcc-2 and lgc-55 are affected equally to the serotonergic pathway genes, while serotonergic pathway genes, such as cat-4, are affected less than some terminal features. However, the authors conclude: "This indicates that HSN partially differentiates, but fails to activate expression of the complete HSN transcriptome". An equally possible conclusion is that they are affecting the differentiation of HSNs, but since they are missing many TFs from their collection, they only affect it partially.

We apologize if our statement was misleading in the initial version of the manuscript. As well pointed by the reviewer, our data shows that these TFs are required for proper expression of a broad range of HSN terminal features and not only the 5HT pathway genes. This finding is very important for the rest of the manuscript as we find these TFs work according to the 'TF collective model' originally proposed by Furlong lab. Accordingly, HSN TF collective defines the HSN neuron regulatory landscape (and not just the 5HT biosynthesis pathway gene expression).

We totally agree with reviewer that other TFs, not analyzed in this manuscript, can produce lineage defects that lead to missing HSN (for example *hlh-14* mutants (Frank et al. 2003). However, this is not the case for the so far identified members of the HSN TF collective. In our single mutant analysis we want to make clear that the lack of expression detected for some markers is not due to the lack of the cell itself but a lack of transcription of that gene. We agree with the reviewer our single mutant analysis shows that HSN differentiation is only partially affected. We thank the reviewer for the remark and we have now modified the text to try to be more precise as follows:

"This indicates that, in each of the analyzed single mutants, HSN is present but shows broad but partial differentiation defects."

- How is HLH3 important for the serotonergic identity when it is not expressed in the adults? The neurons do continue to synthesize serotonin in adult worms. It could be acting as a pioneer factor, or it could be relaying its activity to a downstream transcription factor, but the authors never address this.

We apologize for not being clear enough on the role of HLH-3, as also pointed by reviewer #1. In the revised version of the manuscript we have tried to clarify this issue with additional experiments and re-writing of the section: detailed analysis of *hlh-3* expression by GFP fusion to HLH-3 in the endogenous locus (Figure 2—figure supplement 1A) shows that, post-embryonically, HLH-3 starts being expressed in the HSN at third larval stage and its expression is highest at late third larval stage and early forth larval stage, this expression shortly precedes and overlaps onset of expression of most HSN effector genes (Figure 2—figure supplement 1C). We also demonstrate that *hlh-3* postembryonic expression in *hlh-3* mutants is sufficient to rescue *tph-1* expression defects (Figure 2C). Taking into account that we found functional bHLH binding sites in *tph-1* and *cat-1* CRMs (Figure 5), altogether our data supports the hypothesis that *hlh-3* function is directly required to initiate expression of some HSN effector genes. A role for TFs exclusively in initiation but not maintenance has been previously reported. For example, embryonic *ttx-3* expression in AIY neuron is dependent on the TF REF-2 but its maintenance of expression does not require REF-2 but TTX-3/CEH-10 activity (Bertrand and Hobert 2009). There are also examples in mouse of TFs required for establishment but not maintenance of cell fate (Bettess et al., 2005, Hoogenkamp et al., 2009, Martins et al., 2005) or required for maintenance but not initiation of expression (Galceran et al., 2001, Harms et al., 2014).

HLH-3 belongs to the asc proneural factor family and is ortholog to mouse Ascl1 Figure 2—figure supplement 1D). Ascl1 is expressed in serotonergic progenitors and early postmitotic serotonergic neuroblasts. Besides its role in neuronal progenitors specification it has well-demonstrated roles inducing neuronal differentiation in serotonergic and other neuronal types (Pattyn et al., 2004, Sommer et al., 1995). Ascl1 is also part of the TF combinations used to induce serotonergic fate from human fibroblasts (Vadodaria et al., 2015, Xu et al., 2015). Similar to *hlh-3*, Ascl1 expression is downregulated from mature neurons. Temporal dysregulation of Ascl1 activity by sustained expression or inhibition of degradation leads to neuronal differentiation defects (Imayoshi et al., 2013, Urbán et al., 2016). Similarly, we observed that precocious and sustained expression of *hlh-3* leads to HSN differentiation defects (Figure 2D). *hlh-3* function shows additional shared features to Ascl1 like regulation of expression of panneuronal features (see *rab-3* expression defects Figure 1E) or being an upstream regulator of *egl-46* (Figure 3A) (Ascl1 directly regulates Insm1, ortholog of *egl-46*) (Castro et al., 2011). Finally, Ascl1 can substitute *hlh-3* function in HSN differentiation (Figure 8F). Altogether our new data suggest that in the HSN TF collective *hlh-3* acts as a proneural gene in the induction of HSN terminal fate.

As suggested by reviewer #3 *hlh-3* could be relaying its activity to a downstream TF. Indeed, proneural factors are known to regulate expression of other bHLH TFs that are later required for neuronal differentiation such as NeuroD1 (Guillemot and Hassan 2017). We performed RNAi experiments to knock down the activity of all *C. elegans* bHLH TFs but we fail to detect either *cat-1* or *tph-1* expression defects (See attached graphs at the end of the document). As negative results with RNAi are not conclusive we have decided not to include them in the current version of the manuscript.

Ascl1 has been shown to act as a pioneer factor to facilitate binding of additional TFs (Wapinski et al., 2013). It would be very interesting to address if HLH-3 can also bind closed chromatin. We have not address this issue in the manuscript because we think it is out of the scope of the paper, the main message of the paper is the identification and characterization of HSN TF collective as a whole, our ability to predict HSN enhancers and the phylogenetic conservation. We believe that a more detailed analysis on HLH-3 function as proneural factor is a project on its own for future manuscripts.

This data is now explained in the subsection “AST-1 acts as temporal switch for HSN maturation” and Figure 2 and Figure 2—figure supplement 1.

- In Figure 2E, can the authors really decide on the fate maintenance based on one marker? Moreover, most of the TFs have a very minimally (although significantly) penetrant effect.

We thank the reviewer for this excellent suggestion and agree that additional markers should be added. We have now included *cat-1* expression defects by RNAi (Figure 3C) as well as temperature shift experiments performed with temperature sensitive alleles for *ast-1, sem-4* and *unc-86* (Figure 3—figure supplement 1).

Do they believe that the egg-laying defect is because of this effect? Could it be that this mutant is affecting other cell types that are causing the egg-laying defects?

Mutant alleles for all members of the HSN TF collective show obvious egg-laying defects. As suggested by the reviewer, it is well established that some of the members of the HSN TF collective have pleiotropic actions in other tissues that also contribute to the egg laying phenotype: *sem-4* and *egl-18* are required for VPC specification and vulva formation (Eisenmann and Kim 2000, Koh et al., 2002), *sem-4* is also required for correct specification of vulval and uterine muscles (Basson and Horvitz 1996). HLH-3 is expressed in the VCs and it is required for correct *unc-40* expression (Doonan et al., 2008). Moreover, we have also determined that the expression of other genes in VC4/5 neurons is affected in *hlh-3* mutants. Finally, we have observed that *egl-46* is also expressed in VC neurons. Thus, for most of the HSN TF collective the egg-laying phenotype is not exclusively due to HSN differentiation. In support to this idea we find that our HSN-specific rescue experiments restore HSN gene expression but do not suppress the egg laying phenotype. We thank the reviewer for this remark, and we have now included a comment on this regard in the text:

'Of note, our rescue experiments restore *tph-1* expression but do not rescue egg-laying defects. HSN TF collective have pleiotropic actions in other tissues that also contribute to the egg laying phenotype (Basson and Horvitz 1996, Doonan et al., 2008, Eisenmann and Kim 2000, Koh et al., 2002)what explain the egg laying defects in the HSN specific rescue experiments.'

- As stated earlier, Figure 5 is quite confusing. It shows complex regulatory interactions that cannot be put in a model, as the data are incomplete. Moreover, the authors are testing only TFs with low penetrance and are extrapolating for the whole series.

We thank the reviewer for this remark, we have modified the text to try to clarify our results (subsection “HSN TF collective shows enhancer-context dependent synergistic relationships”). Additionally, we have now expanded our double mutant analysis including more double combinations as well as cis-cis and cis-trans double combinations (Figure 6 and Figure 6—figure supplement 1 and Figure 6—source data 2). We now show synergistic relationships in 9 out of the 15 possible HSN TF collective pair combinations. Synergistic effects include examples for all members of the HSN TF collective and it is the most frequent genetic interaction found in our double mutant analysis (Figure 6—source data 2). Thus, although not all combinations of TFs might be able to synergize we find that all HSN TFs show synergistic relationships and that synergy is a very common feature as would be expected for the TF collective model.

Nevertheless, as stated by reviewer # 3 our data show a complex scenario that cannot be put in a model or used to make predictions or extrapolations, in fact, this is an important message of this part of the manuscript: we find different types of genetic interactions for every reporter gene and for different combinations of TFs pairs indicating there is not a general rule. We apologize if this was misleading, now we have change the text to try to clarify this point. We think this data is very informative. Despite the fact that these TFs act as a collective, they work in such a flexible way that interactions among them are determined specifically by the enhancer context they bind to.

As for the second part of the remark by the reviewer, the use of mutant alleles showing low penetrance is a requirement to be able to unravel genetic interactions. Strong phenotypes would not allow us to detect synergy or additivity.